

# Chronology of thrust propagation from an updated tectono-sedimentary framework of the Miocene molasse (western Alps)

Amir Kalifi[1,2], Philippe-Hervé Leloup[1], Philippe Sorrel[1], Albert Galy[3], François Demory[4], Vincenzo Spina[2], Bastien Huet[2], Frédéric Quillévéré[1], Frédéric Ricciardi[2], Daniel Michoux[2], Kilian Lecacheur[1],
Romain Grime[1], Bernard Pittet[1], Jean-Loup Rubino[2]

[1] Univ Lyon, Univ Lyon 1, ENSL, CNRS, LGL-TPE, F-69622, Villeurbanne, France.
[2] Total SA, CSTJF, Avenue Larribeau, 64000 Pau, France.
[3] CRPG, 15 rue Notre Dames des Pauvres, 54500 Vandœuvre-lès-Nancy, France.
[4] CEREGE, Technopôle de l'Arbois-Méditerranée, BP80, 13545 Aix-en-Provence

*Correspondence to*: Amir Kalifi ([kalifi.amir@gmail.com](mailto:kalifi.amir@gmail.com))

**Abstract.** After more than a century of research, the chronology of the deformation of the external part of the Alpine belt is still controversial for the Miocene epoch. In particular, the poor dating of the foreland basin sedimentary succession hampers
a comprehensive understanding of the kinematics of the deformation. Here we focus on the Miocene Molasse deposits of the northern subalpine massifs, southern Jura, Royans, Bas-Dauphiné, Crest and La Bresse sedimentary basins through a multidisciplinary approach to build a basin-wide tectono-stratigraphic framework. Based on sequence stratigraphy constrained by biostratigraphical, chemostratigraphical (Sr-isotopes) and magnetostratigraphical data between the late Aquitanian (~21 Ma) and the Tortonian (~8.2 Ma), the Miocene Molasse chronostratigraphy is revised with a precision of
~0.5 Ma. The Miocene Molasse sediments encompass four different palaeogeographical domains: (i) the oriental domain, outlined by depositional sequences S1a to S3 (~21 to ~15 Ma), (ii) the median domain characterized by sequences S2 to S5 (~17.8 to ~12 Ma), (iii) the occidental domain, in which sequences S2a to S8 (~17.8 to ~8.2 Ma) were deposited and, (iv) the Bressan domain, where sedimentation is restricted to sequences S6 to S8 (~12 to ~8.2 Ma). A structural and tectono-sedimentary study is conducted based on new field observations and the reappraisal of regional seismic profiles, thereby
allowing the identification of five major faults zones (FZ). The oriental, median and occidental paleogeographical domains are clearly separated by FZ1, FZ2 and FZ3, suggesting strong interactions between tectonics and sedimentation during the Miocene. The evolution in time and space of the paleogeographical domains within a well-constrained structural framework reveals syntectonic deposits and a westward migration of the depocenters, and allows to establish the following chronology of thrust propagation at the western alpine front: (i) A compressive phase (P1) corresponding to thrusting above the
Chartreuse Orientale Thrust (FZ1), which was likely initiated during the Oligocene. This tectonic phase generated reliefs that limited the Miocene transgression to the east; (ii) the ~W-WNW/E-ESE-directed compressive phase (P2) involving the Belledonne basal thrust, which activated the Salève thrust (SAL) fault and successively FZ2 to FZ5 from east to west. Phase P2 deeply shaped the Miocene palaeogeographical evolution and most probably corresponded to a prominent compressive phase at the scale of the Alps during the early to middle Miocene (between 18.05 +/- 0.25 Ma and ~12 Ma). In those ~6 Myr,



the Miocene sea was forced to regress rapidly westwards in response to westward migration of the active thrusts and
exhumation of piggy-back basins atop the fault zones; (iii) the last phase (P3) of Tortonian age (~10 Ma), which likely
implied a significant uplift (350 m minimum) of the Bas-Dauphiné basin, whereas horizontal motions prevailed within the
Jura Mountains.

## 1 Introduction

Foreland basins result from the flexural warping of the lithosphere in response to the orogenic load induced by a continental
collision (Dickinson, 1974; Beaumont, 1981). Along the flexural profile of the foreland basin, the accommodation space
increases progressively towards the orogeny and is maximal near the deformation front (DeCelles and Giles, 1996). As the
deformation front advances, the entire flexural profile is forced to migrate, and additional secondary controls interact with
the flexural migration. Thrust systems are activated synchronously with the sedimentary infill between the deformation front

and the inner part of the orogenic wedge. There, small-scale basins are carried on top of growing thrusts and constitute
"piggy-back basins" (Ori and Friend, 1984). As a result, foreland basin strata (including piggy-back basins, DeCelles and
Giles, 1996) are among the most reliable witness of the geometry and kinematics of growing structures (Suppe et al., 1992)
and thus, of the advance of the deformation front.

The arcuate form of the Western Alps results from a N-NW-directed continental collision during the Eocene and the

Oligocene (Ford and Lickorish, 2004; Dumont et al., 2008, 2011, 2012) and a W-WNW-directed convergence during the
Oligo-Miocene (Butler, 1992a; Dumont et al., 2011, 2008). This evolution was recorded in the foreland basin strata. From
the Oligocene to the Aquitanian, the larger flexural basin was located in the North Alpine Foreland Basin (NAFB, Fig. 1A)
where Molasse deposits are very thick (up to 4 km; Bonnet et al., 2007; Burkhard and Sommaruga, 1998). During the middle
to late Miocene, the NAFB was uplifted (Ford and Lickorish, 2004), and the depocenter migrated to the west along the

western Alpine Foreland basin in France (Allen and Bass, 1993; Lamiraux, 1977) (Fig. 1A). The present study focuses on
the transitional area between these 2 domains (Fig. 1A), which correspond to the southern prolongation of the NAFB
(southern Jura synclines) constituted by lower Miocene strata, and the northern termination of the western Alpine Foreland
basin (Bas-Dauphiné basin) constituted mainly by middle to upper Miocene deposits (Latreille, 1969; Nicolet, 1979;
Kwasniewski et al., 2014). The Bas-Dauphiné basin is located in front of the Vercors-Chartreuse subalpine massifs and

southern Jura synclines (Fig. 1B) where Miocene deposits outcrop in piggy-back basins above NNE-SSW-striking Miocene
thrusts that root in a basement thrust below the External Crystalline Massifs (ECM) (Laubscher, 1992; Bellahsen et al., 2014;
Deville and Chauvière, 2000; Deville et al., 1994, 1992). In the NAFB and the southern Jura synclines, Miocene syntectonic
deposits are dated to the late Burdigalian (e.g. Allen and Bass, 1993; Beck et al., 1998; Deville et al., 1994; Garefalakis and
Schlunegger, 2019) and a westward depocenter migration occurred between the early and the middle Miocene (Lamiraux,

1977; Bass, 1991). Southwards, in the Vercors-Chartreuse and Bas-Dauphiné basin, syn-tectonic deposits are, however, still





insufficiently described. Hence this hampers a comprehensive understanding of the onset, and the chronology, of the Alpine deformation over the Miocene.

This study aims at reappraising the Miocene deformation history of the Western Alpine foreland basin. Based on $^{87}Sr/^{86}Sr$ dating combined with new sedimentological fieldworks, new sequence stratigraphy interpretations for upper Aquitanian to

lower Langhian sedimentary successions outcropping in piggy-back basins of the subalpine massifs and the southern Jura synclines have been proposed (Kalifi et al., 2020). Here, we complete this work with data from the Bas-Dauphiné, Crest and La Bresse basins, and propose an updated chronostratigraphy for the whole area based on integrated biostratigraphical, magnetostratigraphical and new chemiostratigraphical dating of well-logs and sedimentological sections. A new structural analysis is also conducted based on fieldwork data along with a re-interpretation of available seismic lines, geological maps

and published cross-sections. Taken together, the sedimentological, chronostratigraphical and structural approaches enable an updated calendar of the Subalpine massifs and Southern Jura deformations and shortening phases, in response to the Miocene W-WNW directed convergence alpine phase.

## 2 Geological setting

The western Alpine foreland basin corresponds to a part of the peripheral foreland basin of the Cenozoic Alpine orogeny

(Fig. 1A). The Alpine orogeny originates from the closure of the Tethyan Ocean and subsequent continental collision between Eurasia and Adria (or Apulia) (Nicolas et al., 1990; Pfiffner et al., 1997). The study area (Fig. 1B) includes (i) the subalpine massifs (i.e., Vercors, Chartreuse and Bauges massifs) and the French southern Jura, where the Miocene "Molasse" deposits are preserved within synclinal structures and constitute the infill of piggy-back basins; (ii) the Bas-Dauphiné, La Bresse and Crest basins (Fig. 1B), where deposits are poorly deformed and constitute the infill of the foreland

basin. These deposits generally lie unconformably on the thick Mesozoic substratum, or conformably on Oligocene continental deposits (Bass, 1991; Butler, 1992b; Kalifi et al., 2020; Allen and Bass, 1993; Gidon and Arnaud, 1978), and result from the second shallowing-upward cycle of the western alpine foreland basin overfilled phase (Sinclair and Allen, 1992). During this cycle, the regional paleogeography corresponded to a narrow and shallow seaway that connected the Mediterranean Sea (ex-Tethys) to the NAFB (ex-ParaTethys, Allen and Bass, 1993; Bass, 1991; Demarcq, 1970; Rubino et

al., 1990).

In the study area, the Cenozoic chronostratigraphical framework is still poorly constrained and relies mainly on lithostratigraphical subdivisions (Giot, 1944; Pelin, 1965; Bocquet, 1966; Latreille, 1969; Lamiraux, 1977; Mortaz-Djalili, 1977; Mortaz-Djalili and Perriaux, 1979; Nicolet, 1979; Mujito, 1981; Bass, 1991; Allen and Bass, 1993; Berger, 1985, 1992), since biostratigraphical data are scarce. The main compressive phase has been commonly dated to the end of the

Miocene (i.e., Tortonian to Messinian; ~11–5 Ma) (Butler, 1989, 1992; Gidon et al., 1978). However, seismic data from the southern Jura synclines (Deville et al., 1994; Beck et al., 1998), as well as field observations (Blanc, 1991; Kalifi et al.,





2020), suggest that deformation started much earlier with syntectonic sedimentation of the earliest marine deposits (Burdigalian-Langhian; 20–14 Ma).

$^{87}$Sr/$^{86}$Sr and sedimentological analyses (Kalifi et al., 2020) defined a new and well-constrained sequence stratigraphy for the

lower Miocene (with depositional sequences S1, S2 and S3 deposited from the late Aquitanian [~21.3 Ma] to the early Langhian [~16 Ma]) in piggy-back basins (e.g., the subalpine massifs and the southern Jura area). The upper Burdigalian onset of the deformation is evidenced by the sedimentary transition of flexural subsiding distal deposits to syntectonic proximal deposits (Kalifi et al., 2020). This transition corresponds to the progressive migration of the orogenic wedge towards the foreland basin and is materialized by: (i) a shallowing-upward sedimentation defined by shallow-marine

successions capped by gravel-rich fan delta (marine to continental) and continental (riverine) deposits; (ii) growth strata relationships, and (iii) abnormally-thick sediment accumulations induced by an increased subsidence due to the migration of the depocentre located in front of the adjacent thrust belt and their piggy-back basins.

In the study area, the principal structures correspond to cover folds and thrusts, striking NNE-SSW and rooted in the Belledonne basement thrust, which accommodate the last WNW-ESE shortening phase of the Alpine collision wedge

(Mugnier et al., 1990; Doudoux et al., 1982; Bellahsen et al., 2014; Menard and Thouvenot, 1987). These thrusts juxtapose Mesozoic units over Cenozoic or Upper Cretaceous units. Balanced sections of the sedimentary cover in the Vercors and Chartreuse massifs document that horizontal shortening increases from south (6 km) to north (22 km) (Mugnier et al., 1987; Bellahsen et al., 2014; Philippe et al., 1996, 1998). The easternmost major thrust has large, up to 10 km offsets (Deville and Chauvière, 2000). To the north, at the front of the Bauges and Bornes massifs (Fig. 1B), this thrust corresponds to the

*"chevauchement des nappes inférieures"* of Doudoux et al. (1982), and the *"chevauchement frontal des Bauges"* (φB, Fig. 1B) of Gidon (1999). The thrust extends southwards in the Chartreuse massif, and was successively referred to the *"Chevauchement de la Chartreuse orientale"* (Gidon, 1964, 1995) (φOr, Fig. 1B), the *"Central Chartreuse thrust zone"* (Butler and Bowler, 1995), the *"chevauchement subalpin principal"* (Gidon and Arnaud, 1978; Gidon et al., 1978), the *"Subalpine front"* (Deville and Chauvière, 2000), or the *"Chartreuse orientale thrust"* (Philippe et al., 1998). The southern

prolongation of this fault beyond the Isère valley is contentious, but possibly corresponds to the *"Moucherotte thrust"* (φMo, Fig. 1B) (Debelmas, 1965; Gidon, 1981), whose geometry and structural interpretation are still debated (Gignoux and Moret, 1952; Debelmas, 1953, 1966; Gidon, 1981).





**Figure 1: (A) Location of the western Alpine foreland basin and the study area. NAFB= North Alpine foreland basin. (B)**
**Structural map of the subalpine massifs, the southern Jura and adjacent basins (Bas-Dauphiné, Crest, La Bresse). Names of the**
**main faults are indicated within orange circles: 1 to 5 are Fault Zones (FZ) defined in this study; PT=Penninic thrust; Am=**
**Internal Belledonne thrust; E=Entrevernes thrust; SAL = Saleve thrust; J= Jura thrust; SU= Sulens klippen; AN= Annes klippe.**
**Local fault names from the literature: φB= Bauges thrust; φEJu= External Jura thrust; φIJu= Internal Jura thrust; φMo=**
**Moucherotte thrust; φor= Oriental Chartreuse thrust; φvor= Voreppe thrust; φRe= Rencurel thrust; Bur= Buron thrust; V-C=**
**Voiron-Chirens fault. Mountains names: AR= Aravis ; CA= Chailles ; CN= Crête du Nu ; CO= Conest ; EP= Epine ; GC= Grand**
**Colombier ; GM= Grande Moucherolle ; GMI=Grand Manti ; GF= Gros Foug anticline; MC= Montaud col; MD= Monts-d'or ;**
**MM= Monts-du-Matin ; MO= Moucherotte ; NE= Néron ; P= Proveyzieux anticline ; PO= Poliénas ; PR= Pont-en-Royans**
**anticline ; OU= Outherans ; RA= Ratz ; RO= Royans ; RU= Rumilly syncline ; SA= Salève ; SAS= Sassenage ; SE= Semnoz ;**
**TO= Tournier ; VG= Vallée du Grésivaudan ; VU= Vuache. Seismic profiles appear as red lines and wells as red triangles.**

West to this thrust zone, other major thrusts involve Miocene Molasse deposits. In the Vercors massif, from east to west,
these thrusts are: (i) the *"Rencurel thrust"* (φRe, Fig. 1B) or the *"thrust zone 3"* (Watkins et al., 2017). In the South, the
thrust originates at the tip of a NW-SE left-lateral ramp (Barféty et al., 1967; Gidon, 1964), and can be followed to the north
until the Montaud col (MC, Fig. 1B). Its northward prolongation is still disputed: the thrust is connected either to the
northern part of the *"thrust zone 2"* (Watkins et al., 2017) (see below), or to the *"Voreppe thrust"* in the Chartreuse Massif
(φVor, Fig. 1B) (Gidon, 2018; Gidon and Arnaud, 1978); (ii) the tectonic front of the Vercors massif, which partly
corresponds to the *"thrust zone 2"* of Watkins et al. (2017), (iii) the *"thrust zone 1"* (Watkins et al., 2017) at the western
border of the Monts-du-Matin massif (MM, Fig. 1B); (iv) the *"St-Lattier anticline"* (St-La, Fig. 1B) (Deville et al., 1992).

In the Chartreuse massif, at least three thrust zones exist west of the *"Chevauchement de la Chartreuse orientale "* (φOr, Fig.
1B). From east to west they are: (i) the *"Voreppe thrust zone"* (φVor, Fig. 1B) (Butler and Bowler, 1995; Gidon, 1994), also
named the *"Voreppe fault"* (Gidon et al., 1978; Gidon and Arnaud, 1978) or the *"chevauchement φ1 de la Chartreuse*
*occidentale"* (Gidon, 1988; or *"F1"*, Gidon, 1964) from St-Laurent-du-Pont northwards, the prolongation of this accident
becomes unclear (Butler, 1992a); (ii) the Ratz anticline (RA, Fig. 1B) corresponding to the tectonic front of the Chartreuse
massif and, (iii) the *"Voiron-Chirens"* fault (V-C, Fig. 1B) indirectly deduced from a 100m vertical offset of middle
Miocene deposits (Nicolet, 1979).

Further north, west of the Bauges and Bornes massifs, the southern Jura synclines become progressively wider than in the
south. The synclines are separated by anticlines that correspond to the hangingwalls of blind thrust fault systems at depth
(Beck et al., 1998; Doudoux et al., 1982). From west to east, these anticlines constitute: (i) the Salève montain (SA, Fig. 1B)
(Gorin et al., 1993; Mastrangelo and Charollais, 2018) and (ii) the Gros Foug mountain (GF, Fig. 1B). Further west, in the
Jura massifs, three main thrust systems exist from east to west: (i) the thrust involving the Grand-Colombier anticline (GC,
Fig. 1B); (ii) the *"chevauchement interne du Jura"* (φIJu, Fig. 1B) (Philippe, 1995) which corresponds to the northern
prolongation of the thrust involving the Tournier anticline (TO, Fig. 1B) and, (iii) the *"chevauchement externe du Jura"*
(φEJu, Fig. 1B) (Philippe, 1995).

At the latitude of Chambéry, the transition between widely spaced and more closely spaced thrusts and folds (Fig. 1B) has
been attributed to the presence of a more efficient décollement level in the Triassic layers in the north (Philippe, 1995;
Philippe et al., 1996).



As mentioned above, the detailed chronology of these thrusts is poorly constrained. Moreover, Miocene deposits were poorly investigated in the Vercors and Chartreuse massifs. As a result, the main compressive phase could have started during the late Miocene (Gidon et al., 1978; Gidon (www.geol-alp.com); Butler, 1989b, 1989a, 1992b, 1992a). However, this is off-phase with the observations to the north, in the southern Jura, where detailled investigation of lower Miocene deposits in

seismic profiles and field observations revealed seismites and growth-strata relationships which suggests that a compressive phase started during the early Miocene (Deville et al., 1994; Beck et al., 1998; Blanc, 1991; Rangheard et al., 1990).

## 3 Material and methods

Sedimentological and stratigraphical analyses were conducted from 35 well-outcropping sections of the Miocene Molasse deposits (sections 4, 5, 13, 16, 22 are presented in Kalifi et al., 2020), and from partially preserved sections (<40m)

outcropping in adjacent localities. Sedimentary successions, up to 1050 m-thick, were logged at the decimeter (dm) to meter (m) scale in the field. The combined analyses of textural characteristics, clastic and biogenic components, bed thickness, bed organization and geometry, sedimentary structures and paleocurrent measurements allow the definition of 25 facies grouped into 11 facies associations (FA) (Kalifi et al., 2020). Depositional sequences were identified based on FA evolution and the main stratigraphical surfaces (Embry, 1993, 1995), as detailed in Kalifi et al. (2020). Depositional sequences were confirmed

and some only identified, using Posamentier and Allen (1999) methodology on spontaneous potential (SP) and gamma-ray logs (GR) data from 28 well-logs located in the Bas-Dauphiné basin.

Three dating approaches have been combined in order to constrain the ages of the depositional sequences. The results have been calibrated on the GTS2016 chronostratigraphic chart (Ogg et al., 2016). Biostratigraphical dating was conducted on field and well-log samples using calcareous nannofossils (calibrated using Young et al., 2017), dinoflagellate cysts (using

Hardenbol et al. (1999) biozonation and calibrated using TimeScale Creator 7.4), and foraminifera (calibrated using BouDagher-Fadel (2015), Lirer et al. (2019) and Wade et al. (2011) biozonations; see Kalifi, 2020 for details). Calcareous nannofossils, foraminifera and mammal dating available in the literature were also implemented (21 mammal localities were calibrated on the GTS 2016 using Time scale Creator 7.4, Kalifi, 2020 for details). Strontium (Sr) isotopes dating was performed on samples collected on the field, as well as on a few well-log samples. The Sr isotope ratios were measured on

marine carbonate skeletons (oysters and pectens) at CRPG (Centre de Recherches Pétrographiques et Géochimiques) in Nancy. A thorough inspection of shells was preliminarily conducted at TOTAL using (i) cathodoluminescence, in order to select shells yielding a pristine structure as well as to avoid recrystallized structures; (ii) and/or stable isotope ratios ($\delta^{13}$C and $\delta^{18}$O), in order to evaluate possible diagenetic disturbance (Hudson, 1977; Nelson and Smith, 1996; Hudson, 1975). The maximum confidence zone corresponds to $\delta^{13}$C values comprised between + 3.5 ‰ and – 1 ‰, as the global ocean $\delta^{13}$C

during Miocene was comprised between 0 and + 2.5 ‰ (Hayes et al., 1999) and an additional margin of about +/- 1 ‰ must be applied for epicontinental seas (Saltzmann and Thomas, 2012). Samples with $\delta^{13}$C values lower than -5 ‰ are excluded as they are considered as diagenetized (Kalifi, 2020). Corresponding ages are derived from measured $^{87}$Sr/$^{86}$Sr ratios using



the LOWESS non-parametric regression curve of McArthur et al. (2012). A mean value was calculated where more than one sample was available for one stratigraphic level. The average of the $^{87}Sr/^{86}Sr$ ratios was converted to ages using the

LOWESS 5 table (McArthur et al., 2012). The average uncertainty corresponds to the largest value between the values defined by 2 standard errors or s.e. (= 2 x the standard deviation of isotopic ratios divided by the square root of the number of data involved) and the individual minimum error of the sample (taking into account the measurement error and the standard error). More details about the geochemical analyses used in this study are given in Kalifi (2020) and Kalifi et al. (2020).

Paleomagnetic results were obtained for 84 samples. 33 and 51 oriented cores or blocks were retrieved from the Forezan (24 layers) and Grésy-sur-Aix (48 layers) sedimentary sections 4 and 5. Samples were subjected to stepwise alternating field (AF) demagnetization, thermal demagnetization or a composite procedure (Thermal up to 350°C and then AF) of the Natural Remanent Magnetization (NRM). All remanent magnetizations were measured using the Superconducting Rock Magnetometer SRM 760R (2G Enterprises) of the CEREGE (Aix-en-Provence, France). AF online and thermal experiments

were performed using the magnetically-shielded oven MMTD80 (Magnetic Measurements Ltd.). All paleomagnetic results (directions, treatment and statistics) are archived in Table S1. According to the moderate quality of the paleomagnetic results (especially for the Grésy-sur-Aix section) and the rather low values of bedding dips, no fold test could be done. The magnetic component most resistant to the demagnetization protocols was considered to be (or be close to) the directional geomagnetic signal acquired during just after deposition of magnetic particles.

The structural study was carried out by a systematic reappraisal of published structural maps and cross-sections at the light of new field data. Published geological maps at 1:50000 scale (BRGM), as well as local structural observations and cross-sections (i.e. *http://www.geol-alp.com* and references therein) were used together with new field observations (stratification and fault measurements, observation of panoramas, study of stratigraphic successions) in key areas at more than 730 locations (Fig. S1) to determine the geometry of the structures and their continuity. Seismic profiles (Fig. 1B) that partially

penetrate the front of the subalpine chains are located on the western edge of the Vercors and Chartreuse massifs (91CHA1-2, 91VER1, 82SE01-91VER2, Deville et al., 1992; Deville, 2021) and on the western edge of the Bauges and Bornes massifs (88SV01, 88SV02-HR528, 88SV03, 88SV05, 88SV06, 88SV07-HR535; Gorin et al., 1993; Signer and Gorin, 1995; Beck et al., 1998; available at BRGM) and were reinterpreted by using the stratigraphic column presented in Figure 1 and by integrating new field data. Further information has been obtained from well-logs in the southern Jura synclines, the La

Bresse and Bas Dauphiné basins (Fig. 1B). The combined analysis of these data allows us to propose a new structural map of the area (Fig. 1B) along with four regional cross-sections (A to D). The cross-sections B and C in the Chartreuse massif were balanced and restored backward to the timing of deformation provided by this study, using the "flexural slip algorithm" of the Move software. In order to ensure realistic fault geometries, the restoration was performed using the top of the most laterally continuous pre-thrusting unit (i.e. Barremian-Aptian limestones).



## 4 Results and interpretations

### 4.1 Dating

#### 4.1.1 Chemostratigraphy

129 Sr dating were conducted in this study (Table S2). 12 samples display evidence for diagenetic alteration with $\delta^{13}$C values < -5 ‰ (Table S2) and cathodoluminescence imagery exhibiting recrystallization patterns. Four samples were further interpreted as reworked samples, and considered as outliers (Table S2) since the stable isotopes data exhibit no evidence of diagenetic bias ($\delta^{13}$C >-5‰) but the obtained ages are older than the age of the stratigraphic level. Excluding those 16 recrystallized or outliers samples, the $^{87}$Sr/$^{86}$Sr ratios range between 0.708338 and 0.708914 corresponding to stratigraphic ages between 21.45 +/- 0.30 Ma (late Aquitanian) and 8.58 +/- 1.78 Ma (Tortonian), thereby allowing to date 113 stratigraphic levels.

#### 4.1.2 Magnetostratigraphy

For the Forezan and Grésy-sur-Aix sections (respectively sections 5 and 4, Fig. 2; locations in Fig. 3), the NRM intensity ranges between 1.53 $10^{-5}$ and 8.43 $10^{-3}$ Am$^{-1}$ with an average of 1.12 $10^{-3}$ Am$^{-1}$, and between 3.19 $10^{-5}$ and 3.95 $10^{-3}$ Am$^{-1}$ with an average of 8.83 $10^{-4}$ Am$^{-4}$, respectively. The Forezan section (enriched in clayey facies) presents slightly higher NRM intensities and better paleomagnetic results probably due to finer and more abundant ferromagnetic grains. Orthogonal projection of paleomagnetic results exhibits different patterns: (i) a stable reverse polarity slightly overprinted by a normal polarity (Fig. S2A) in many samples of the Forezan sedimentary section, (ii) a large normal overprint with a path toward reverse polarity (Fig. S2B) for many samples of the Grésy sedimentary section, (iii) rarely stable normal polarity (Fig. S2C) attributed to full normal remagnetization, or eventually a subchron, and (iv) in the 868-870m interval of the Grésy section (for 2 neighboring samples), a stable normal polarity affected by a post-lock tilting (Fig. S2D). Declination and inclination of the stable magnetization, or the direction displayed in the very last steps of demagnetization, were plotted together with the reversal angle (Fig. S3A, B). Due to large normal magnetic overprints, the reversal angle, which is the angle between the calculated direction and the expected direction for the normal geomagnetic polarity, helps to determine geomagnetic polarity scale especially in carbonates affected by partial but intense remagnetizations (e.g. Demory et al., 2011). The entire Forezan sedimentary section displays very high reversal angles (Fig. S3A) attributed to the record of the reverse geomagnetic Chron C5Cr according to the age frame inferred from Sr dating (Fig. 2). Most of the Grésy-sur-Aix section is characterized by reverse polarities except from the normal polarity recorded at 772 m, which may be attributed to Subchron C5Dr.1n., and tilted normal polarities (Fig. S2D and Fig. S3B) located at 868 m - just below a major unconformity – that may be related to the record of Chron C5Dn (Fig. 2). In the case of post-lock tilting, the reversal angle is higher than expected for normal polarity.





**Figure 2: Magnetostratigraphy of the Forézan and Grésy-sur-Aix sections, integrated with Strontium and biostratigraphic dating.**

### 4.1.3 Biostratigraphy

34 out of 74 analyzed samples enabled the biostratigraphical dating of sedimentary sections based on nannofossil assemblages (Table S3). Species that allowed precise stratigraphical calibrations are *Coccolithus miopelagicus* (NN5–NN8;





260 14.91–10.55 Ma), *Helicosphaera ampliaperta* (middle part of NN2–top NN4; 20.4–14.91 Ma), *Helicosphaera scissura* (upper NN2–NN5; 20.1–14.91 Ma), *Helicosphaera stalis* (NN6–NN11; 13.53–5.59 Ma), *Sphenolithus belemnos* (base NN3–top NN3; 19–18 Ma), *Sphenolithus heteromorphus* (base NN4–top NN5; 18–13.5 Ma).

21 out of 34 analyzed samples allowed biostratigraphical constraints from dinoflagellate cyst assemblages (Table S4). Species that allowed precise stratigraphic calibrations are *Cousteaudinium sp.* (upper D16–top D17; 20.5–14.8 Ma),

265 *Ectosphaeropsis burdigalensis* (lower D16–lower D18; 23.2–13.8 Ma), *Hystrichosphaeropsis obscura* (lower D17–top D19 ;18.65–7.5 Ma), *Systematophora placacantha* (Eocene–D18; Eocene–13 Ma).

Only 2 out of 40 analyzed samples allocated a biostratigraphical dating of sedimentary sections based on foraminiferal assemblages (Table S5). In addition, 13 foraminiferal assemblages were already available from the literature (Lamiraux, 1977; Latreille, 1969; Mein, 1985; Aguilar et al., 2004) and were implemented in this study using recent biostratigraphical

270 charts (Table S5). The species allowing firm stratigraphical constraints are *Globigerinoides sicanus* (MMi4a–MMi4c; 16.1–14.87 Ma sensu Lirer et al., 2019), *Paragloborotalia bella* and *Praeorbulina circularis* (respectively N5b–N9; 20.4–14.0 Ma and N8b–N9a; 15.9–14.5 Ma; sensu BouDagher-Fadel, 2015).

21 mammal assemblages available from the literature (Fig. S4; see Kalifi (2020) and references therein for more details) also revealed to be useful to constrain the age of the sea retreat (appendix A) between Zones MN6 and MN9 (15.2–9.5 Ma).

The Figure S4 compiles a synthesis of the Miocene biostratigraphy chart of the study area.

**4.2 Sequence stratigraphy**

Absolute and relative dating calibrations were crucial to constrain the timing of the sequence stratigraphy interpretations arising from the 35 sedimentological sections and 28 well log data (Fig. 3 for location; see also Kalifi (2020) for sedimentological and stratigraphical detail) integrated in this study. To add more timing constrains on these sections and

well-logs, 84 dating calibrations obtained from partially preserved successions outcropping in adjacent localities complete here the previously published data (Kalifi et al., 2020; Fig. 3, appendix A). These additional points derive from new field observations and data from the literature (Kalifi, 2020 and references therein). Taken together, the whole dataset allowed the establishment of 10 marine depositional sequences between the late Aquitanian (S1a) and the early Tortonian (S7), capped by a continental upper Tortonian sequence (S8) (Fig. 4A). The Sr-based ages obtained in the basal transgressive deposits

coincide with eustatic sea-level rises (Fig. 4B, C), suggesting that the depositional sequences are eustatically-driven (see also Kalifi et al., 2020), except for S8 (Fig. 4A, B). Twelve palaeogeographical zones (A to L, Fig. 3) with distinct infill histories were identified based on their specific sedimentological and chronostratigraphical records (appendix A). The error bars of the Sr-based ages obtained in the basal transgressive deposits do not allow the identification of diachronic transgressions from one paleogeographic zone to the other (Fig. 4C). Therefore, we assume that each transgression occurred sub-

synchroneously at the scale of the study area, and that the eustatic sea-level rise age (Fig. 4B) will be used to: (i) constrain in time depositional sequences without dating controls and/or, (ii) homogenize the age of the sequence at the basin scale.



**Figure 3: The 12 palaeogeographical zones of the Miocene basin of the subalpine massifs, southern Jura and adjacent basins (Bas-Dauphiné, La Bresse, Crest). Sedimentological and chronostratigraphical records of each zone are provided in appendix A.**





**Figure 4: The Miocene chronostratigraphy of the subalpine massifs, southern Jura and adjacent basins (Bas-Dauphiné, La Bresse, Crest). (A) Synthetic section of the Miocene of the study area and the 11 Miocene depositional sequences. Legend in Fig. 5; (B) The eustatic curve of Miller et al. (2005) recalibrated with the Global Time Scale (GTS) of Ogg et al. (2016); (C) Average Sr-based ages of the transgressions in the palaeogeographical zones. n: number of age values. Only ages of samples with $\delta^{13}C$ values >-1 ‰ (Table S2) are considered here (maximum confidence zone).**

The 12 Miocene palaeogeographical zones can be grouped in four domains based on our chronostratigraphy (Fig. 5):

**(i) Oriental domain**: this depositional area corresponds to the Rumilly-Chambéry synclines (Zone A) and the Proveyzieux-Lans synclines (Zone B). Located in the easternmost part of the subalpine chains, they are characterized by the occurrence of the lower marine Burdigalian sequences (S1a, S1b). The upper Burdigalian (S2a, S2b) is also very thick (~800 m in the zone A) and is characterized by distal marine deposits in the North (zone A) and coarse-grained deltaic deposits in the South (zone B) (Fig. 5). The S3 (+S4?) is on the other hand very poorly developed (<50 m), and corresponds to coarse-grained deltaic to continental deposits. The middle Miocene and younger deposits are absent in this domain.



**Figure 5: Spatio-temporal distribution of the Miocene depositional sequences of the sub-alpine massifs, southern Jura synclines, Bas-Dauphiné, Crest and La Bresse, which highlight the four depositional domains resulting from the sequence stratigraphical interpretation of the 12 palaeogeographical zones (presented in details in appendix A).**

**(ii) Median domain**: from north to south, this depositional domain encompasses the Novalaise (zone C), Voreppe (zone D) and Rencurel-Méaudre synclines (zone E). The lower Burdigalian (S1a, S1b) is absent in this domain set in the westernmost position of the subalpine chains (Fig. 5). The first deposit corresponds to the marine upper Burdigalian (S2a, S2b), which is about 100 to 200 m thick. Sequence 3, on the other hand, is much thicker in this domain (~400 m in zone C). The S4 and S5 depositional sequences of the middle Miocene do not exceed 150 m in thickness and are characterized by coarse-grained deltaic to continental deposits (Fig. 5). The upper Miocene (above S5?) is in turn rarely recorded, thin or not expressed at all.

**(iii) Occidental domain**: This domain concerns the Tour-du-Pin area (H), Bièvres region (F), Bonnevaux plateau (J), Chambaran plateau (K), Royans syncline (G) and Crest basin (L). These areas are characterized by the absence of the lower





Burdigalian (S1a, S1b) and the discontinuous occurrence of the upper Burdigalian (S2a, S2b). The Langhian and Serravalian depositional sequences (S3, S4, S5, S6), however, are remarkably well developed (Fig. 5). The thickest upper Miocene deposits (S7, S8), mainly containing continental sediments, are also found in this domain (Fig. 5).

    **(iv) "Bressan" domain**: this domain concerns solely the Bresse basin (I), where lower and middle Miocene (S1 to S5) marine sediments are absent (Fig. 5). This zone was only flooded during the major eustatic transgression of the S6 sequence.

Upper Miocene deposits (S7 and S8) are particularly thick in this domain and mainly made of continental products.

## 4.3 Overall structure and main thrust zones

Structural maps and cross-sections have been reappraised with the aim to produce a new regional structural map (Fig. 1B). West of the Belledonne external crystalline range, the Mesozoic cover is affected by numerous thrusts, nearly all diping to the east. The uppermost thrust sheet is the Penninic thrust outcroping at the base of the Sulens and Les Annes klippen (Fig.

1B). This thrust roots east of Belledonne and was active during the Oligocene (Simon-Labric et al., 2009; Dumont et al., 2008, 2011, 2012; Doudoux et al., 1982). To the West and structurally below the Penninic thrust, five main NNE-SSW striking faults zones, locally offset by NE-SW right-lateral faults, straddle the Vercors and Chartreuse massifs. We describe below these fault zones in more detail, from East to West. We also evaluate the continuity of the fault zones between the southern area (Vercors and Chartreuse massifs) where folds and thrusts are closely spaced, and the northern area where folds

and thrusts are more widely spaced.

### 4.3.1 "Chartreuse orientale thrust" or Fault Zone 1 (FZ1)

Within the Chartreuse massif, a major thrust bringing Lower Cretaceous units on top of younger Cretaceous units has long been distinguished in the literature. It runs from Chambéry to the west of Grenoble and was named differently (Gidon, 1964; Butler and Bowler, 1995; Gidon and Arnaud, 1978; Gidon et al., 1978; Deville and Chauvière, 2000; Philippe et al., 1998).

Here we refer to this thrust as the Chartreuse orientale thrust (φOr, Fig. 1B). It was associated with a large tectonic displacement of about ten kilometres (Deville and Chauvière, 2000) at the latitude of Chambéry. Along its southern portion this thrust brings the Néron syncline affecting "Urgonian" (=Barremian-Aptian) limestones on top of the Proveyzieux syncline filled with Miocene deposits. In its central part, it brings Mesozoic sediments on top of the Charmant Som anticline (CS, Fig. 1B), while further north Cretaceous strata are found over Oligocene deposits (Doudoux et al., 1992a, b; Gidon and

Barféty, 1969). The Chartreuse orientale thrust is affected by numerous right-lateral NE-SW faults (Fig. 1B). North of Chambéry, the thrust connects with the western front of the Bornes and Bauges massifs characterized by Mesozoic units thrusted over Oligocene and Miocene deposits (φB, Fig. 1B).

The prolongation of this thrust towards the south across the Isère valley is more contentious. The valley has been sometimes interpreted as the location of either a saddle of the folds (e.g. Gignoux and Moret, 1952) or possibly a fault (Gidon and

Arnaud, 1978) that would offset the structures outcropping on both sides. The most recent detailed study concluded that there is no offset in the downstream part of the valley, but possibly a tear fault in the upstream (Gidon, 1995). In that



interpretation, the St-Martin thrust has no counterpart across the Isère valley, but only discontinuous faults such as the Dent-du-Loup and Pas-du-Mortier thrusts, and most structures show an apparent dextral offset across the valley that is only due to a small deflection of their direction as they are more widely spaced in the Vercors (south) massif than in the Chartreuse

(North) massif (Fig. 7A). South of the valley, the Moucherotte Thrust that brings Cretacous sediments atop Miocene of the Villard-de-Lans syncline has a complex 3D geometry, connecting with the poorly exposed Perrières Fault through the Bruzier right lateral fault (Debelmas, 1965; Gidon, 2020b) (Fig. 7A). New field observations (Fig. 6) combined with the published geological maps and publications suggest a new structural map, corresponding cross-sections (Fig. 7) and interpretation. In fact, North of the Isère valley, the Proveyzieux syncline is a more complex structure since it corresponds to

two synclines separated by a pinched anticline (the Proveyzieux anticline in Fig. 6A; Fig. 7A). South of the Isère valley, the Sassenage fold is also an anticline partly capped by Miocene deposits (Fig. 6B). Either side of the Isère valley, the two anticlines have the same axis direction, trending ~N20° in the Proveyzieux anticline and ~N15° in the Sassenage anticline (Fig. 7A). In both cases, the eastern limb of the anticline is overthrusted by Cretaceous sediments including "Urgonian" limestones (Moucherotte thrust in the south, Néron thrust in the north) strongly suggesting that the thrust system is continous

across the Isère valley. Therefore, we propose that the Proveyzieux and Sassenage anticlines correspond to the same structure. The precise trace of the thrust is unclear, hidden by Quaternary sediments, but the Moucherotte – Bruzier – Perrieres thrust appears to be offset from the Neron thrust by ~2km along a left-lateral NW-SE strike-slip fault along the Isère valley (Fig. 7A). Such a fault is compatible with the general direction of shortening. A major thrust thus appears to be continuous from the western front of the Bauges massif to the western front of the Moucherotte range (from north to south

φB, φOr, φMo), and is referred here as the Fault Zone 1 (FZ1) (Fig. 1B).











**Figure 6: Field and aircraft pictures of the Chartreuse Orientale and Moucherotte thrusts (FZ1) in southern Chartreuse and northern Vercors. m= Miocene; Cs= Upper Cretaceous; Urg. = Urgonian (Barremian-Aptian limestones); Ci= Lower Cretaceous; Jc= Upper Jurassic limestones; Jm= Upper Jurassic marls. Dashed white lines underline the stratifications. Thrust faults in red. Geographical and geological locations of viewpoints A to I in Fig. 7A. (A, B) The Néron and Moucherotte thrusts, aerial views from the N. The Proveyzieux Miocene deposits are underthrusted below the Néron syncline. Beyond the Isère valley, the Sassenage fold capped with Miocene deposits is underthrusted below the Moucherotte massif. Note that Proveyzieux syncline corresponds to two synclines separated by an anticline. (C) View towards the south of the Col de l'Arc and the Crête des Crocs: the folded Urgonian is thrusted above the Upper Cretaceous. (D) Urgonian limestones of the Moucherotte anticline in the FZ1 hangingwall. (E) View towards the west of the eastern flank of the Moucherotte massif with the FZ1 geometry. (F) View towards the north of the Cornafion syncline. The Urgonian is overturned, in the FZ1 footwall. (G) Duplication of the Jurassic series in the Pieu klippen, view from the North; (H) Zoom on the anticline on the hangingwall of the FZ1 in the Pieu klippen. (I) View towards the north of the Grande Moucherolle, where autochthonous series are weakly deformed.**





Fig. 7: Structure of the Chartreuse orientale thrust (FZ1) in northern Vercors and southern Chartreuse. See Fig. 1 for legend. (A) Structural map. Fig. 6 viewpoints A to I are located. (B-G) geological cross-sections 1 to 6 (see location in A).

Although the westward Moucherotte thrust has long been described, its detailed geometry is still discussed. We conducted a detailed mapping based on field observations at 210 stations combined with a detailed analysis of previous maps and publications and one aircraft flight, in order to constrain the 3D geometry of the Moucherotte range geology. On the western flank of the Moucherotte range, the Cretaceous beds of the Moucherotte anticline are thrusted above the Miocene on the top of the Sassenage anticline (Fig. 6B and Fig. 7D) (Gidon, 1981). The vertically dipping Cretaceous series pass farther east, on





the eastern flank of the Moucherotte massif, to Lower Cretaceous strata gently dipping to the west above Upper Jurassic limestones (Comboire range). The marls outcropping at the foot of the Comboire range did not deposited during the Late

Jurassic as previously mapped on the Vif geological map (Barféty et al., 1967). They rather deposited during the Early Cretaceous (based on the occurrence of *Berriasella*; Gidon, 2020a), which implies a duplication of the above-described series and the emergence of a west-dipping thrust. This thrust most probably corresponds to the Moucherotte thrust, making the Moucherotte range a klippen (Fig. 7D).

A few kilometers further south, near the "Col de l'Arc", the Cretaceous serie is also duplicated. To the west, Urgonian beds

of the Moucherotte anticline are juxtaposed on top of the Upper Cretaceous (Fig. 6C, Fig. 7E). The hanging wall structure is complex with a syncline of Upper Cretaceous sediments in the west and a boxed anticline overlyied by a monocline in the east (Fig. 6D). To the east of the Col de l'Arc, below the "Crête des Crocs", the Lower Cretaceous exhibits steep east-directed dips (~70°, Fig. 6C), and is allochthonous above more gently dipping Urgonian limestones (Fig. 6E). The top of the Moucherotte range is thus a klippen that can be traced ~1 km farther to the south to the "Pierre Vivari" (Fig. 7A). Further

south, the Cornafion syncline trends N10° and shows an inverted eastern limb (Fig. 6E, F) in the thrust footwall (Fig. 7F). At this location, the Moucherotte klippen has been completely eroded. On the eastern flank of the Moucherotte massif, the thrust can be traced at the top of the autochtonous Urgonian limestones (Fig. 6E, Barféty et al., 1967), to the base of the Comboire range (Fig. 7A), confirming that the upper part of the massif constitutes a klippen.

South of the Moucherotte massif, and West of Vif, the Upper Jurassic and Lower Cretaceous series are duplicated,

constituting the Pieu klippen (Fig. 6G; Fig. 7F) (Barféty et al., 1967). At the western edge of this klippen, Upper Jurassic limestone allochthonous strata form a bent-fault anticline (Fig. 6H) above the autochthonous Jurassic and Cretaceous strata (Fig. 7F) indicating thrusting to the west. We interpret this klippen as the southern/lower extension of the Moucherotte one (Fig. 7F). Further south, the Mesozoic formations are weakly deformed, locally exhibiting west-verging anticlines such as the "Grande Moucherolle" (Fig. 6I, Fig. 7G). The Moucherotte klippen did probably not extand south of the Double Brêche

fault, south of which the footwall syncline is no more observed (Fig. 7A).

The geometry of the southern extension of the Chartreuse orientale – Moucherotte thrust (FZ1) in the Vercors massif, which exhibits a persistent vergence to the west, but dip to the west, to the east of the Pieu klippen and Comboire range (Fig. 7D, F), strongly suggests that a folding phase post-date the thrust. No other evidence of this thrust is found east of the Pieu klippen suggesting that it roots east of the Ecoutoux - Conest anticlinorium, which is the southwestern prolongation of the

external Belledonne range (Fig. 7A, F; Fig. 1B). From south to North, the apparent offsets are measured at 1.1 km along the Pieu section (Fig. 7F), 3.3 km along the Pic St-Michel and Moucherotte sections (Fig. 7E, D) based on the base of the Barremian Aptian unit, and 2.8 km along the Neron section based on the base of the Lower Cretaceous unit (Fig. 7B).

The eastern emergence of the Moucherotte Thrust reaches Comboire (Fig. 7A). Near Grenoble at the south-eastern edge of the Chartreuse, the "Corenc-Jalla" folded-thrust system (Blanchet and Chagny, 1923; Gignoux and Moret, 1952; "φc" in

geological map of Grenoble; Gidon and Arnaud, 1978) probably corresponds to the northern prolongation of the Moucherotte thrust (Gidon, 1981) (Fig. 7A, B). This thrust system has been affected by a late folding that also affected the





autochthonous units (Blanchet and Chagny, 1923; Gidon, 1981) (Corenc; Fig. 7B). Barfety and Gidon (1996) further considered that this fault, that they termed "φ1", is a major thrust which re-emerges to the east of the Isère valley, on the western edge of Belledonne massif (Fig. B1A, D). There, along the Grésivaudan valley on top of the Belledonne basement,

φ1 is sub-parallel to the Mesozoic series it affects, diping 20-40° to the west, and duplicating the Dogger (Barfety and Gidon, 1996) (Fig. B1A, D). φ1 thus appears as a top to the west fault that is a tilted thrust (Deville et al., 1994), whose offset probably increases northwards, as inferred from the deformation intensity of disharmonic folds (Barfety and Gidon, 1996). Further to the north, at the eastern edge of the Bornes massif, the Dogger and Lias are tightly folded in a top-to-the-west shear zone (Barféty and Barbier, 1983; Barfety and Gidon, 1996; Doudoux et al., 1982, 1999) that also corresponds to a

thrust tilted by the uplift of the Belledonne range (Doudoux et al., 1999) (Fig. B1A, B, C).

The Chartreuse Orientale thrust (FZ1) thus appears to run continously from the western front of the Bornes to the west Moucherotte thrust in the south, and from the east of Moucherotte to east of the Bornes, defining a ~120km long thrust nappe (Fig. 1B). This thrust sheet was termed the Aravis-Mt Granier unit (Bellahsen et al., 2014) or the Subalpine nappe (Pfiffner, 2014). These authors further suggested that this thrust roots in the basement east to the Belledonne massif, which would have

been uplifted after motion along the thrust ended (also see Lacassin et al., 1990; Menard and Thouvenot, 1987).

**4.3.2 Voreppe thrust or Fault Zone 2 (FZ2)**

In the Vercors massif, the main major thrust west of the FZ1 is the "Rencurel thrust" (φRe, Fig. 1B; Fig. 7A) which outcrops near "La-Balme-Rencurel" (Fig. 8A). There, the thrust exhibits Barremian-Aptian limestones thrusted above overturned Miocene Molasse deposits (Fig. 8B). The apparent offset for the top of the Urgonian is of 1.5 km along this section (Fig.

8C). Miocene strata are limestones bearing Chlamys praescabriusculus, which are upper Burdigalian in age (Fig. A5). At this location, the fault strikes N3, 40°E with slickensides having a pitch of 80°S. Analysis of the dips in the Lower Cretaceous marls indicates that the hangingwall of the fault is characterized by two anticlinal structures separated by a syncline (Fig. 8C). Further east, two other thrusts, with smaller offsets (0.8 and 0.9 km), are observed parallel to the Rencurel thrust (Fig. 8C). According to the published geological map (Barféty et al., 1967), the thrust extends southwards for 6 km until it

branches at the tip of a NW-SE left-lateral fault (Fig. 1B). Towards the north, the thrust is continuous until the "Col de Montaud" (MC, Fig. 7A), with Miocene deposits observed in the footwall. At this location, it was proposed that the thrust connects with the thrust at the western front of the Vercors (Watkins et al., 2017). However, the fault does not cut across the footwall Miocene deposits (Gidon and Arnaud, 1978), but more likely veers from N-S to NNE-SSW to reach Veurey-Voroize where Upper Jurassic sediments are found thrusting atop the Miocene (Fig. 1B; Fig. 7A). N-E of the Isère valley, we

suggest that the "Rencurel thrust" prolongates to the "Voreppe thrust" across the Isère valley (Fig. 7A). Indeed, the seismic profile 91VER1 that follows the Isère valley (Fig. 9A) reveals a major thrust at this location (Fig. 9C) with a 1.5 km apparent offset of the basal Lower Cretaceous reflector. Hence, we interpret the Rencurel and Voreppe thrusts as the same tectonic structure.



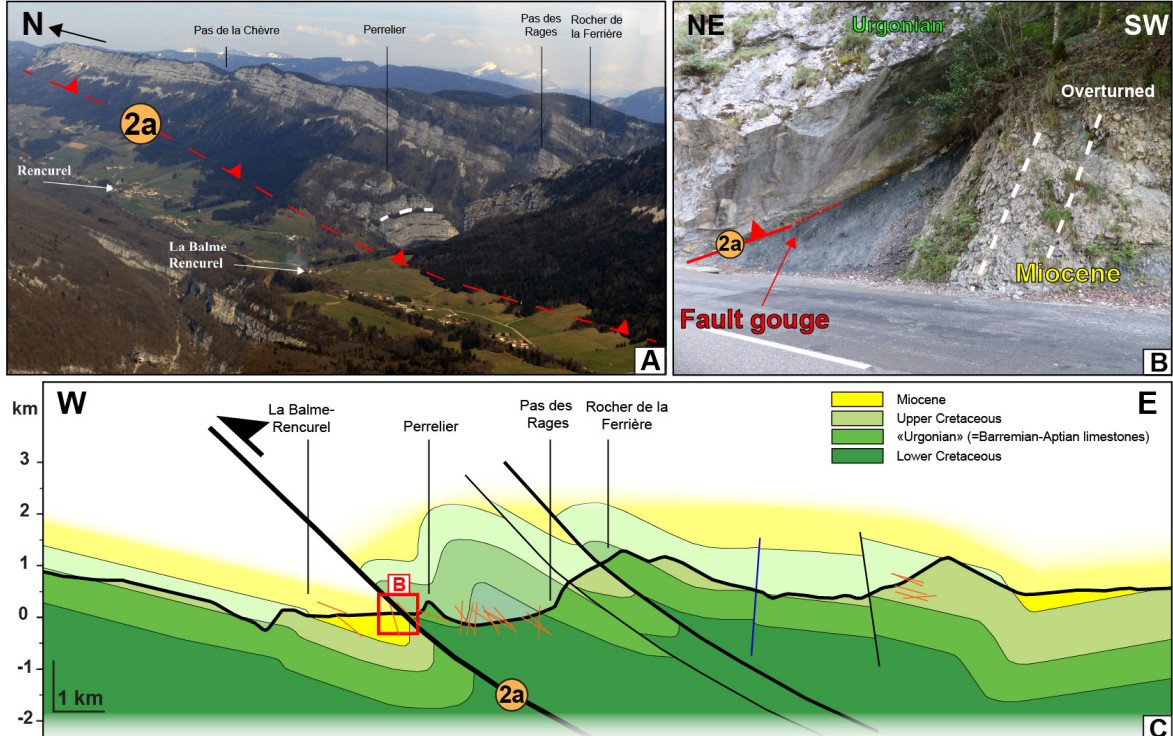

**Figure 8: The Rencurel thrust in the Vercors massif. Orange lines= projected dip measurements. Blue line = Right-lateral strike slip fault. Black line = Thrust. (A) The N-S Rencurel thrust that juxtaposes Cretaceous units on top of Miocene Molasse deposits. (B) Outcrop of the Rencurel thrust along the D103 road, SE to the La Balme-Rencurel village, exhibiting Barremian-Aptian limestones thrusting on top of overturned Miocene deposits striking N160, 70°E. (C) E-W cross-section along the "gorges de la Bourne". See Fig. 7A for location of the cross-section and dip measures on map view.**

Further north, the Voreppe thrust is mapped as a continuous structure bringing Jurassic sediments on top of Miocene deposits of the Voreppe syncline (Fig. 1B) (Gidon and Arnaud, 1978). North of St Laurent-Du-Pont, this thrust terminates, but other thrusts continue further to the north: to the west two faults merge together before bounding the western limb of the l'Epine anticline (EP, Fig. 1B), while to the east a thrust bound the Miocene deposits from the steep westen flanck of the Outherans (OU) anticline (Fig. 1B). The seismic profile 91CHA1 stands across the four faults (Fig. 9A, B). Its interpretation shows that the two western faults (2b and 2c) merge at depth just above the basement, and that they could also merge with the Voreppe thrust (2a) just east of the eastern limit of the seismic profile (Fig. 8B). We thus interpret the western faults [2b and 2c] as the northern prolongation of the Voreppe thrust [2a], with a 8 km long left-lateral step over (Fig. 1B). Within the step over, short NW-SE left-lateral faults are reported on the 1/50 000 map (Gidon, 1970a) (Fig. 1B) that may concur to the westward shift of the shortening from the Voreppe thrust to the l'Epine thrust. Based on the basal Lower Cretaceous reflector, the apparent offsets are 2.2 km for the Voreppe thrust [2a], 2.3 km for the fault [2b] and 1.1 km for the fault [2c] (Fig. 9B). Following this interpretation, the FZ2 thrust system would be shifted westwards, and continues northward as the [2b] thrust bordering the western flank of the "Epine" [EP], "Grand Colombier " [GC] and "Crêt du Nu " anticlines (Fig. 1B).



Figure 9: Seismic profiles along the Vercors and Chartreuse massifs (seismic lines published by Deville [2021]). (A) Geographical and geological locations of the interpreted seismic lines. (B) Seismic profiles 91CHA1 and 91CHA2. Interpretation is proposed using PA-1 well data and fieldwork data. (C) Seismic profiles 91VER1. Interpretation is proposed using BR-1 well data and fieldwork data. (D) Seismic profiles 82SE01-91VER2. Interpretation is proposed using SLF-2, SL-2 and extrapolated BB1 and GVA-1 well data and fieldwork data.

The seismic profile 91CHA1 shows that the Outeran thrust [OU] is almost parallel to the three other ones and has a 1.3 km apparent offset (Fig. 9B). To the south, the Outherans (OU) anticline corresponds to the Chartreuse median anticline of Gidon (1964, 1990; CS, Fig. 1B, Fig. 10A), while to the North it most likely connects with the Gros Foug (GF) anticline across the Chambéry valley (Fig. 1B). The geometry of the Gros Foug (GF) anticline can be observed on seismic profiles 88SV01 and 88SV03 (Fig. 10C, D), and supports a west-dipping Gros Foug (GF) thrust with an apparent offset between 2 and 2.5 km of the basal Upper Jurassic limestones reflector. This thrust appears to root in the Triassic detachment level above the basement (Fig. 10C, D) and is probably the northern extent of the Outherans thrust in between the Charteuse Orientale (FZ1) and the Voreppe thrust (FZ2) (Fig. 10A).





The seismic profiles also reveal another thrust fault between FZ1 and FZ2: the Salève (SAL) thrust bounding the Salève (SA) anticline (Fig. 1B, Fig. 10A, B) that also roots above the basement in the Triassic. The apparent offset of SAL fault is

evaluated to be 2.5 km based on the basal Barremian-Aptian limestones reflector in seismic profile 88SV01 (Fig. 10D), and on the basal Upper Jurassic limestones reflector in seismic profile SV03 (Fig. 10C). The seismic profiles only allow two hypotheses for the geometrical relationship between FZ1 and the SAL faults: (i) the FZ1 and the SAL faults are connected at depth (Deville and Sassi, 2006; Beck et al., 1998); (ii) the FZ1 and the SAL faults are distinct (Le Guellec et al., 1990; Doudoux et al., 1982).

**Fig. 10: The salève (SAL) and Gros Foug (GF) faults between the Chartreuse massif and the Jura synclines (A, legend in Fig. 1) as based on seismic profiles 88SV07+HR-535 (B), 88SV03 (C) and 88SV01 (D).**






### 4.3.3 The Royans - Ratz thrust or Fault Zone 3 (FZ3)

The western edge of the Vercors and Chartreuse massifs corresponds to open anticlines characterized by steep west-facing
forelimbs and gently dipping back-limbs. Such anticlines are continuous from Pont-en-Royans in the Vercors massif (PR) to
the Chailles anticline in the Chartreuse massif (CA), with the Ratz (RA) anticline in an intermediate position (Fig. 1B).
Seismic profiles crossing these structures (Fig. 9A) reveal that these anticlines formed in the hanging wall of a major thrust
zone: the Royans - Ratz thrust (FZ3). This thrust zone corresponds to a single fault bringing Barremian-Aptian to Upper
Jurassic limestones above Miocene strata. According to the basal Upper Jurassic limestones reflector, the apparent offset is
evaluated to be 1.5 km along the 91CHA1-2 and 91VER1 seismic profiles (Fig. 9B, C). Alternatively, the seismic profile
82SE01 (Fig. 9D) shows that the main fault (3) has an apparent offset of 4.5 km, but that other minor faults also occur. The
FZ3 roots in the Triassic detachment level above the basement (Fig. 9).

### 4.3.4 The Monts du matin – Voiron thrust or Fault Zone 4 (FZ4)

To the west of FZ3, seismic profiles reveal a blind thrust system, with two thrusts likely connected at depth, here referred as
the FZ4, with a total apparent offset of the basal Upper Jurassic limestones reflector of ~100-300 m in the north (91CHA1-2;
91VER1, Fig. 9B, C) and ~650 m in the south (82SE01, Fig. 9D). To the south, this thrust system bounds the Mont-du-
Matins anticline (MM, Fig. 9A). The thrusts crosscuts inherited normal faults (i.e. the Saint-Nazaire fault [SN], Fig. 9D) as
already reported by Deville et al. (1992), and roots in the Triassic detachment level (Fig. 9).

### 4.3.5 The St Lattier thrust or Fault Zone 5 (FZ5)

West of the FZ4, the seismic profile 82SE01 (Fig. 9D) highlights the presence of an additional blind thrust system that
corresponds to the St-Lattier anticline (St-La, Fig. 1B) (Deville et al., 1992), here referred as the FZ5. The extension of FZ5
to the north is not well constrained, but may coincide with thrusts exhibiting small offsets (<100 m) that are rooted on an
inherited normal fault seen along profile 91VER1 ~40 km further north (Fig. 9C). FZ5 is not observed on the seismic profile
91CHA1-2, in the north (Fig. 9B).

### 525  4.3.6 Summary of the structure of the subalpine and South Jura domains and regional cross sections

Five main west verging thrust zones have been indentified that all trend NNE-SSW on an updated structural map of the
subalpine ranges (Fig. 1B) and presented on four regional cross sections (Fig. 11). Folds in between the thrusts have the
same general trend and the cross-sections have been drawn in a WNW-ESE direction.
More specifically, sub-surface data and surface data (field observations and geological maps at 1:50 000) were used to build
cross-section A west of FZ1, while east of FZ1, the construction of the cross-section relies on the analysis of 1:50 000 scale
geological maps and the ECORS-CROP profile (Le Guellec et al., 1990; Roure et al., 1990; Mugnier and Marthelot, 1991)
recently revised by Pfiffner (2014). The Cenozoic filling consists of Oligocene and Eocene deposits and is involved in





superposed nappes (Doudoux et al., 1982). The Upper Jurassic limestones is duplicated three times according to the BZN1 borehole further north (Fig. 1B for location) (Charollais and Jamet, 1990). Seismic imaging implies the presence of an
uplifted basement compartment (Le Guellec et al., 1990). To the east, we have also integrated the Montjoie valley synthetic section of Gidon (2019) (Fig. B1A, B, C) for the Aravis and Belledonne chains relationships. In order to build the westernmost part of section B up to the FZ1, sub-surface data and surface data (field observations and geological maps at 1:50 000) were used. Between FZ1 and the summit of the Grand Manti (Fig. 11B), the construction of the cross-section relies on detailed analyses of 1:50 000 geological maps. To the east of the Grand Manti, the data originate from the
northernmost section available in Barfety and Gidon (1996). The results presented in Fig. 6, Fig. 7 and in Barfety and Gidon (1996) (Fig. B1A, D) were used to build sections C and D (Fig. 10C, D) east of FZ2, while sub-surface data (Fig. 9C, D) and results presented in Fig. 7 (for section D) were used to the west of FZ2. Sections B and C have been balanced and restored using the Move software.

The FZ1, has been refolded and roots east of external Belledonne. In this interpretation, large parts of the subalpine ranges
(Moucherotte, East Chartreuse, Beauges, Bornes) constitute klippen. West of FZ1, FZ2 (2a, 2b, 2c) runs from South to North from the Vercors to the l'Epine and the Grand Colombier – Crêt du Nu. FZ2 is interpreted to branch at the top of the basement on a flat decollement in the Triassic evaporites that roots further east below external Belledonne in the Basal Belledonne thrust (Doudoux et al., 1982; Bellahsen et al., 2014, 2012; Butler, 2017; Deville and Chauvière, 2000; Deville et al., 1992, 1994). FZ1 and FZ2 are only 4 to 9 km apart in the Vercors-Chartreuse, but 39 km apart in the Jura-Bauges (Fig.
1B). Such widening is the combination of a fan-like map-shape of the folds and offsets by left-leteral NW-SE faults. The greater distance between the thrust and the inner part of the belt in the Jura is likely to represent a more efficient décollement in the North, where Triassic evaporitic series are thicker (Philippe et al., 1996). Anticlines are found in FZ2 hangingwall and thus in FZ1 footwall (Fig. 1B, Fig. 11), but related to the activity of FZ2 according to Le Guellec et al. (1990) and Doudoux et al. (1982). FZ3 is continuous from the Vercors to the western Chartreuse. Further north, according to published geological
maps, it appears to veer to the west and merge with the "chevauchement majeur du Jura interne" (φIJu, Fig.1B) (Philippe, 1995). The map pattern is similar to that of the FZ2 and interpreted in the same way. As there is no significant basement culmination between FZ2 and FZ3, FZ3 is interpreted to follow the same décollement level and to root underneath the Belledonne external massif as FZ2. FZ4 is only documented from seismic profiles and its apparent offset decreases toward the north. The general fault pattern might suggest that FZ4 could connect with the "chevauchement majeur du Jura externe"
(φEJu, Fig.1B) through the NW-SE Jura transfert zone followed by the Rhône river (J, Fig. 1B) (Philippe, 1995). FZ5 is only documented west of the Vercors. As there is no significant basement culmination between FZ5 and FZ2, FZ5 and FZ4 are interpreted to follow the same décollement level and to root underneath the Belledonne external massif. East verging thrusts are locally described. East of the Royans (RO, Fig. 1B) in the Vercors and in the crêt du Nu in the Jura (CN, Fig. 1B), they appear to be linked with the underlying west verging decollement (Philippe, 1995). In Grenoble (Ecoutoux anticline, Fig.
7A, B; Fig. 11C), they appear as related to a late deformatrion phase (Barfety and Gidon, 1996).





As described above, the NW-SE (N145°) left-lateral South Jura transfert zone appear to offset FZ3 and possibly FZ4. The step over between the Voreppe (2a) and the Epine (2b) thrusts stands in the prolongation of this zone and shows small NW-SE left-lateral faults (Fig.1B) suggesting that it is linked to the same process. Other similar faults are only found at the southern tip of the Rencurel thrust (2a, N140°), and possibly along part of the Isère valley near Grenoble offsetting FZ1 (Fig.

1B). Numerous NE-SW (~N65°) right-lateral faults affect the southern Bauges, the Chartreuse and the northern Vercors (Fig. 1B). These faults offset FZ1 and possibly FZ2 but not FZ3 (Fig. 1B). Both the left-lateral and the right-lateral faults are compatible with an ESE-WNW compression as the main trusts and folds.



**Figure 11: Reginal cross-sections of the study area. A) Cross section location on the structural map. PT= Penninic thrust. AF=**

**Arcalod fault. B) cross section A through the eastern Jura and the Bornes. C) Cross section B through central Chartreuse and**




Belledonne. D) Cross section C through southern Chartreuse and Belledonne. E) Cross section D through Vercors and southern Belledonne.

### 4.4 Relationships between sedimentation and tectonics

#### 4.4.1 The Rumilly syncline

At the footwall of the FZ1, in the Rumilly syncline located to the north of the oriental domain (Zone A, Fig. 5), the Oligo-Miocene succession starts with sub-concordant onlaps on the Mesozoic substratum, according to seismic profiles (Fig. 10). The thickness of the Oligo-Aquitanian continental deposits decreases westward (Fig. 10C, D). Indeed, these deposits are ~200 m thick west of the Rumilly syncline (Fig. 9C, D) (Enay et al., 1970; Gidon, 1970b), while to the east, they reach 1838 m between the footwall of the FZ1 and the hangingwall of the SAL fault (SLV2 well data, Fig. 10A, B), and 1716 m at the

footwall of the SAL fault (SV-101 well data, Fig. 10A, D). Thus, on both sides of the SAL fault, which appears west of the FZ1 (Fig. 10B), the thickness of Oligocene deposits is relatively similar (Fig. 10B, D), suggesting that the Oligo-Aquitanian depocenter was located at the footwall of FZ1. This, along with sub-concordant Oligocene strata on the Mesozoic substratum, suggests that the SAL fault likely activated after the deposition of Oligo-Miocene sediments. This is further consistent with field analyses revealing that the basal Miocene Molasse deposits (upper Aquitanian-lower Burdigalian) are

conformably lying on continental Aquitanian deposits dipping vertically in the eastern flank of the Rumilly syncline (Alby-sur-Chéran, 3, Fig. 3). Altogether, this implies that both Oligocene and Miocene sediments were deformed by the SAL fault after deposition, then after the early Burdigalian.

#### 4.4.2 Grésy-sur-Aix section

To the south of the Rumilly syncline, in the Grésy-sur-Aix sedimentological section (4, Fig. 3), (Fig. 12A) the S2a sequence

basal boundary (dated to 18.05 +/- 0.3 Ma based on correlation with the Forezan section (5), see Fig. 2; Fig. 3 for location) was extrapolated along-strike to the north. In the 88SV01 and 88SV03 seismic profiles (Fig. 9D, C), this major stratigraphic surface coincides with a high amplitude reflector. This reflector is characterized by low-angle toplaps and onlaps that either suggest: (i) an angular unconformity, which would materializes the onset of a tectonic phase; or (ii) an erosive surface above the underlying deposits, which corresponds to a classic feature for sequences boundaries.

The Grésy-sur-Aix section outlines three examples of syntectonic sedimentation within Sequence S2a (Fig. 12). Firstly, S2a is abnormally thick with ~750 m of sediments deposited over ~0.85 Myr (18.05 +/- 0.25 to 17.2 +/- 0.15 Ma, but excluding an age older than 17.23 Ma for top S2a which was recorded during the Chron C5Cr (Fig. 12A); average sedimentation rate of 0.72 +/- 0.32 mm y$^{-1}$), compared to the underlying sequences S1a-S1b, only 200 m thick and deposited over ~3.5 Myr (21.45 +/- 0.3 to 18.05 +/- 0.25 Ma; average sedimentation rate of 0.06 +/- 0.01 mm y$^{-1}$) (Fig. 12A). This suggests a

significant increase of accommodation space at the initiation of S2a, likely related to a regional tectonic event. Secondly, a 15m-thick seismite layer (Fig. 12A, C, D), also found in section 5 (at ~380 m, Fig. 2), is regarded as a regional stratigraphic level that also argue for a major tectonic event. Thirdly, the occurrence of an angular unconformity observed in tidal flat




deposits (Fig. 12A, B), as characterized by a decrease in bedding dip from 18° to 7° up-section and an erosive surface exhibiting westward-directed onlaps above it. This angular unconformity argues for a compressive tectonic event.







**Figure 12: (A)** Grésy-sur-Aix sedimentological section with S2a syntectonic deposits indicated: seismite near the bottom (see also C and D) and angular uncoformity near the top (see also B). The S2a sequence boundary was observed and dated in the Forezan section (5, Fig. 2). Legend in Fig. 5. Black box= Average sedimentation rate of the sequence; **(B)** The angular unconformity has established between 17.53 and 17.23 Ma within Chron C5Dn. White dashed lines = bedding. Red line = erosive surface. Red square = Dip measure; **(C)** Seismite disturbing 15 m of sediments in S2a, with plurimetric blocks tilted in situ and slump balls; **(D)** Detail of an intensively disturbed sedimentary level.

The timing of the angular unconformity is further well-constrained by magnetostratigraphy, and has been associated to Chron C5Dn at 17.35 +/- 0.15 Ma (Fig. 2). Thus, S2a most probably records the onset of a compressive phase. In this interpretation, in the 88SV01 and 88SV03 seismic profiles (Fig.9D, C), the S2a sequence boundary characterized by low-angle toplaps and onlaps is a tectonic-driven angular unconformity, and thus, dates the initiation of compression at 18.05 +/- 0.25 Ma.

### 4.4.3 Southern Jura synclines

Strontium, biostratigraphical and magnetostratigraphical dating were applied on sedimentological sections 3, 4, 5 (oriental domain, zone A) and 13 (median domain, zone C) from the southern Jura synclines immediately north of Chambery (Fig. 3). Sequence stratigraphy interpretations enable E-W correlations in lower Miocene deposits between section 3 (Fig. 13A) and section 13 (Fig. 13D), by using sequence boundaries and maximum flooding surfaces. The thicknesses of S1a-S1b do not exceed 200 m in sections 3, 4 and 5 (Fig. 13A, B, C), and are even absent to the west, in section 13 (Fig. 13D). During S2a, the maximum thickness is ~750 m and is recorded in section 4 (Fig. 13B), while to the west, in section 13 (Fig. 13D), S2a is 145 m-thick. Furthermore, in section 4 (Fig. 13B), the average sedimentation rate increase brutally from 0.06 +/- 0.01 mm y$^{-1}$ during S1a-S1b, to 0.72 +/- 0.32 mm y$^{-1}$ during S2a (Fig. 12A). This firmly demonstrates that a depocenter localized close to section 4 (Fig. 13B) appeared during S2a. Subsequently, the thickest accumulation of the following sequence (S2b) lies further west, at the Forezan locality (275 m, Fig. 13C), thereby suggesting a westward migration of the depocenter between sequences S2a and S2b. This is also consistent with a progressive decrease of accommodation rates from ca. 1.17 mm yr$^{-1}$ to 0.17 mm yr$^{-1}$ between S2a and S2b in section 4 to the east (Fig. 13B). Finally, the maximum thickness of S3 is recorded west of the section 5 (Fig. 13C), at section 13 (Fig. 13D). There, the S3 is 390 m-thick, while in the Rumilly-Chambéry synclines area (Fig. 13), the S3 was never recorded thicker. Furthermore, during deposition of S3, lateral variation of the thickness is associated with significant lateral facies variation. At Loisieux (Fig. 13D, Fig. A3), S3 is mainly characterized by distal marine deposits, while it exhibits coarse-grained deltaic deposits to the east (Fig. 13B, C; Fig. A1). These proximal deposits suggest that the Rumilly-Chambéry synclines area was progressively exhumed, which is also suggested by the continuation of the progressive decrease of accumulation rates recorded between S2a and S2b. Thus, the S3 depocenter is most-probably located in the Loisieux locality (Fig. 13D), further highlighting the westward migration of the depocenter (Fig. 13).







**Figure 13: Westward migration of the depocenter during the lower Miocene (red curved arrow). The sedimentological sections were projected on a E-W axis, which corresponds to a direction orthogonal to the thrusts. See Fig. 3 for location of the transect and sedimentological sections. Red squares = sedimentological sections quoted in the text and used for correlations. The dashed red line corresponds to the angular unconformity presented in Fig. 12B.**

### 4.4.4 Western Chartreuse

In the 91CHA1-2 seismic profile (Fig. 14A), the Miocene sequences were calibrated using the PA-1 well-log (Fig. 14B, C). The thicknesses of S4 and S5 increase from the hangingwall to the footwall of the FZ4, which is likely associated with a decrease in bedding dip up-section (Fig. 14A) suggesting growth-strata in response to FZ4 activity. On the other hand, the underlying S3 deposits are isopacheous both in the footwall and the hangingwall of FZ4 giving an initiation of FZ4 after the deposition of S3. The end of tectonic activity on FZ4 is not so well constrained since seismic data do not allow a detailed observation of S6 deposits, although S6 apparently seals (conformably) the underlying deposits, that might suggest the end





of tectonic activity. This could be in accordance with the fact that the FZ4 offsets solely S4 deposits, while S5 deposits are
655 simply folded that probably indicates a progressive decrease in the tectonic activity from S4 to S5. Alltogether, these
observations suggest that the FZ4 activity was recorded by the S4 and S5 sequences (~15.0 to ~12.0 Ma).

Further east, beyond the artefact located at the emergence of FZ3 (Massieu, Fig. 14A), the identification of S4 is proposed
based on similar seismic facies (Fig. 14D). In the hangingwall of FZ3, the basal Miocene deposits are conformably lying on
the Mesozoic bedrock (on top of the Chailles anticline). This is consistent with fieldwork observations made 5 km to the
660 north, in the Chailles gorges (Fig. 14A, C), where the basal Miocene Molasse deposits have been attributed to sequence S2a
(Fig. 14E) suggesting that FZ3 was inactive during deposition of the S2a. On the FZ3 hangingwall, the overlying sequences
(S2b? -S3) geometries are observed on the 91CHA1-2 seismic profile (Fig. 14A), and their thicknesses decrease gradually
when approaching the Chaille anticline together with a decrease in bedding dip. In the up section, (possibly S2b) S3 and S4
sequences are also characterized by eastward-directed onlaps on the folded S2a deposits (Fig. 14A), thus suggesting growth-
665 strata related to FZ3 activity, between these sequences. Downstream to the Chailles valley (Fig. 14C), above the S2a
sequence, high in the cliff, growth strata (Fig. 14F) is again suggested based on apparent eastward-directed onlaps onto S2a
deposits, in agreement with the interpretation of the seismic profile. However, these outcropping onlaps does not allow to
identifiy more precisely the timing of the growth strata.







**Figure 14: Growth strata relationships inferred for FZ3 and FZ4 based on the 91CHA1 and 91CHA2 seismic profiles, sequential interpretation of the PA-1 well and fieldwork observations. The ages in red boxes correspond to the homogenized ages of the eustatic-driven transgressions (Fig. 4B). (A) Seismic profiles 91CHA1 and 91CHA2 depicting the Miocene reflectors and geometrical relationships associated with FZ3 and FZ4. The red double curved arrows point the growth stratas. Legend in Fig. 9. (B) Sequential interpretation of the PA-1 well and conversions of sedimentary thicknesses into seismic velocities (Deville et al., 1992). Legend in Fig. 15. (C) Geographical and geological locations. Legend in Fig. 1. (D) Zoom on the Massieu area (see location on A) showing similar seismic facies on both sides of the Massieu artefact and suggesting positive offset across FZ3. (E) Tilted concordant Urgonian-Miocene basal conglomeratic contact (S2a). (F) Growth-strata relationships at approximately 20 and 30 m above the river level, corresponding to S2b or S3 deposits according to regional correlation lines.**

### 4.4.5 Bas Dauphiné basin

In the Bas-Dauphiné basin (Fig. 1B), the Miocene Molasse deposits are poorly deformed and only the upper part of the Miocene succession crops out. So, the three SE-NW transects (Fig. 3) presented here are mainly based on the interpretation of borehole data using sequence stratigraphy (Fig. 15) and our updated chronostratigraphy. Correlations are proposed by using the main sequence boundaries elevation (in m.a.s.l.= meters above sea level).

The Bas-Dauphiné basin is separated into two tectonic zones. West of the Montmiral high (Fig. 15B, C), no compressive structures were found (Kalifi, 2020; Couëffé and Tourlière, 2008). On the other hand, FZ4 and FZ5 blind thrusts are found to the east of the Montmiral and L'île Cremieu highs (Fig. 15A, B, C). In this area, the S2a, S2b and S3 sequences exhibit very few variation in thickness (~100 m for S2a-b between VAF-1 and VAF-2 wells, Fig. 15B, and ~50-70 m for S3 between SLF-1, MO-1, MO-2 and MO-3 wells, Fig. 15C), despite the presence of the FZ5 thrusts within transect C and possibly B. On the western edge of the Bas-Dauphiné basin, the absence of S2a-S2b deposits (to the northwest of PA-1, VAF-2 and MO-3 wells, Fig. 15A, B, C) and the thickness variations of S3 to the west of the Montmiral high (Fig. 15B, C) are attributed to a complex inherited paleo-topography (Kalifi, 2020).

East of the Montmiral high, significant variations in depositional thickness are observed for S4 with a maximum located in the footwall of the FZ4. In the transect A (Fig. 15 A), a S4 maximum thickness of 215 m is recorded in the PA-1 well, while to the west, the S4 thickness decreases progressively, reaching 53 m in the CH-1 well. In the transect B, S4 maximum thickness is recorded in the VAF-1 well, with a thickness of at least 161 m (since its top is eroded), while to the west, S4 is 118 m-thick in the BRI-1 well, above the Montmiral high (Fig. 15B). In the transect C, S4 maximum thickness is recorded in the SLF-1 well, with a minimum of 190 m (eroded top), while to the west, S4 is 129 m and 116 m, respectively, in the MO-2 and MO-3 wells (Fig. 15C).

The sequence S5a-S5b also exhibits variation in thickness. In the transect A, a S5a-S5b maximum thickness of 210 m is recorded in the PA-1 well, while it decreases progressively to the west reaching 82 m in the CH-1 well (Fig. 15A). In the transect B, S5a-b maximum thickness is difficult to estimate because the top of the VAF-1 and VAF-2 successions are eroded to the east of the Montmiral high (Fig. 15B). However, a probable depocenter is located to the west of the FZ5, in the FA-1 well, with a thickness of 153 m (Fig. 15B). In the transect C, S5a-S5b depocenter can be located between the Montmiral high and the FZ5, in the MO-2 well (Fig. 15C). There, the S5a-S5b is 204 m-thick, while to the west, it is ~165





m-thick above the Montmiral high (MO-2 and MO-3 wells, Fig. 15C). To the west, the interpretation of the DP-108 well is difficult as only lithological data are available, but might suggest another depocenter with a thickness of 205 m (Fig. 15C).

**Figure 15: Spatio-temporal distribution of Miocene Molasse deposits along SE-NW transects of the Bas-Dauphiné basin. Correlations are proposed based on sequence stratigraphy interpretations using stacking pattern methodology on SP, GR and lithological data from well logs.**



No significant thickness variations are observed for the S6 sequence, which is ~105m-thick along the transect A (Fig. 15A), ~145 m-thick along the transect B (Fig. 15B), and ~70 m-thick along the transect C (Fig. 15C). The variation in thickness for S7 and S8 cannot be estimated as they are rarely present in well-logs (Fig. 15).

### 4.4.6 Bresse basin

To the east, the La Bresse basin is bordered by the l'île Cremieu high, in the south, and by the N-S striking Jura thrusts in the North (Fig. 1B). In the La Bresse basin, the lower Miocene is absent and the area is finally flooded at ~12.0 Ma, only during S6 sequence (Fig. A9). The S6 sequence (~12.0 to ~10.8 Ma, Fig. 4B) and the transgressive deposits of the S7 (~10.8 to ~10 Ma, Fig. 4B) are characterized by marine deposits, while the regressive tract of the S7 and the S8 exhibit purely continental deposits implying a major regression starting during S7.

At the Jura front, the BY-101 well-log (Fig. 3) shows Mesozoic units thrusted atop Miocene deposits (Dumont, 1983). The same Miocene succession outcrops at the eastern border of the La Bresse basin, at Jujurieux (Dumont, 1983; I2, Fig. 3), and corresponds to marine deposits of S6 and S7 based on Sr ages (Table S2), which are consistent with the magneto-biostratigraphical results of Aguilar et al. (2004). Therefore, the S6 and S7 are thrusted by the Jura thrust, at the BY-101 locality. Thus, the onset of the Jura thrust is either synchronous, or posterior, to sequences S6-S7 marine deposits (~12 to 725    ~10 Ma).

At the Jura front, the continental deposits, belonging to the S8 sequence according to mammal-based datations (I4, I5, I6 localities; Fig. 3), could also be partly deposited during the S7 regressive tract, and are characterized by a 200 m-thick continental clay and marl succession (Demarcq, 1970). This important thickness may suggest the existence of a tectonic-controlled depocenter in the footwall of the Jura thrust.

## 5 Discussion

### 5.1 Genetic relationships between the stratigraphical domains and the fault zones (FZ)

The different depositional sequences depict a conspicuous spatial organization at the regional scale defining four depositional domains (Fig. 5). The Oriental and Median domains correspond to narrow longitudinal north-south bands, which are orthogonal to the general ESE-WNW direction of shortening linked to the alpine compression. The age of the 735    basal sequence becomes younger toward the west, indicating a westward migration of the depocenters (Fig. 4C). A similar migration can be deduced from the thickness variation of the sequences (e.g., S2a-S2b, S3 and S4) (Fig. 5). The Oriental, Median and Occidental domains are respectively bounded to the east by the FZ1, FZ2 and FZ3 (Fig. 16). Such agreement between the localisation of the sedimentologic and the structural boundaries suggests a genetic relationship between thrusting, accommodation space creation and sediment infill.

The presence of angular unconformities and seismites within Burdigalian deposits of the southern Jura synclines (Fig. 1B) were already mentioned by Beck et al. (1998), Blanc (1991), and Deville et al. (1994), although the chronostratigraphic





framework was still sketchy at that time. The present study, integrating evidences for syntectonic deposits, growth strata geometries and depocenter migration at the basin scale within a well-constrained chronostratigraphy, allows to depict the Miocene tectono-sedimentary evolution as follow.

**Figure 16: Summary of the structural, chronostratigraphical and tectonostratigraphic results of this study. The 11 sedimentary sequences revealed 12 paleogeographical zones, themselves grouped into 4 palaeogeographical domains. 5 fault zones (FZ1-5) were identified. Based on the established tectono-sedimentary evolution, three compressive phases were identified. The main phase corresponds to Phase 2, which involved the progressive onset, from east to west, of the SAL fault to FZ5 in response to the onset of the Belledonne basal thrust.**

### 5.1.1 The FZ1 activity (Phase 1)

The westward thinning of the Oligocene and Miocene S1a-S1b succession observed in the Rumilly syncline implies a regular subsiding area between ~29 +/- 2 Ma (Rigassi, 1957; base of the Oligocene deposits dated by palynostratigraphy westwards of Annecy, locality 3; Fig. 10A, C) and ~18.05+/-0.15 Ma (S2a sequence boundary, Fig. 10A, C). Moreover, the





westward-directed onlaps on the Mesozoic substratum, as well as the younger age of the base of the Oligocene sediments at the west of the Rumilly syncline (~23 Ma based on mammals, Gaudant et al., 2002, locality 7; Fig. 10A, C), suggest a regular and slow westward migration of the depocenter. In a foreland basin, this geometry is consistent with a foredeep depozone (DeCelles and Giles, 1996) located between the poorly subsiding proximal flank of the forebulge and the footwall of the active (tectonic) orogenic front, where the maximum of subsidence is recorded in response to the orogenic load. The

progressive decrease of depositional thicknesses towards the west might imply the proximity to the forebulge, whereas the maximum depositional thickness recorded to the east of the Rumilly syncline suggests the proximity to the orogenic front. This interpretation is consistent with that of Deville et al. (1994), who previously interpreted the Oligocene sedimentary succession in this area as deposited in a passive foreland flexural basin. The tectonic front is therefore located further east: FZ1, or the Entrevernes thrust (E, Fig. 1B), or even further east. Since marine Miocene deposits have never been described

east of FZ1, we believe that the front zone already formed a morphostructural barrier during the first marine transgression (S1a; ~21.0 Ma, Fig. 4B, C) and resulted in the Oligocene flexural subsidence recorded in the Rumilly syncline. However, Oligocene growth-strata arguments are still needed to confirm such hypothesis.

### 5.1.2 Onset of the SAL, GF and FZ2 faults (beginning of Phase 2)

Southwest of the Rumilly syncline, the E-W correlation between sedimentological sections 3, 4, 5 and 13 (Fig. 3) highlights

important thickness variations (Fig. 13), leading us to propose the following chronology. At 18.05 +/- 0.25 Ma, sequence S2a exhibits a prominent acceleration in accumulation rates especially at the Gresy-sur-Aix locality (Fig. 13B), suggesting the rapid production of accommodation space probably linked to a regional tectonic event. Interestingly, the occurrence of seismites involving a major earthquake is observed within the S2a transgressive tract (Fig. 12). Seismic profiles from the Rumilly syncline (Fig. 10) outline that the S2a basal sequence boundary corresponds to high-amplitude reflectors

characterized by low-angle S1 toplaps and S2 onlaps, thereby advocating for an angular unconformity. Together, this suggests a tectonic event associated with the activation of the SAL thrust at 18.05 +/- 0.25 Ma, since the S2a depocenter is located at the footwall of this fault (Fig. 17). Subsequently, the angular unconformity highlighted in tidal flat deposits, and associated to Chron C5Dn (17.53-17.23 Ma, Fig. 12A, B) recovered at the top of the S2a regressive tract, most probably reflects another regional tectonic event with the depocenter abruptly migrating westwards at that time, at the footwall of the

GF fault (Fig. 13C). These two lines of evidences, therefore, support the hypothesis of the onset of the GF fault during Chron C5Dn (Fig. 17). At ~16.2 Ma (S3), the depocenter migrated westwards again and lies at the footwall of FZ2 (Fig. 13D). There, during deposition of S3, thick distal marine deposits were sedimented, whereas coarse-grained deltaic deposits prevailed to the east of FZ2 (Fig. 13B, C), suggesting a more proximal and uplifted area. Thus, FZ2 was activated during deposition of S3 (Fig. 17).


**Figure 17: Schematic framework summarizing the main phases of thrust propagation at the western alpine front, and its chronology, based on a thorough revision of the regional chronostratigraphy between the Oligocene and the late Miocene. The horizontal and vertical scales are not respected. MC= Massif Central; P= Tectonic phase; T= Transgressive deposits; R= Regressive deposits. The colored boxes on the left side correspond to the homogenized ages of the eustatic-driven depositional sequences (Fig. 4A, B).**







### 5.1.3 Onset of FZ3, FZ4 and FZ5 (ending of Phase 2)

At the hangingwall of FZ3, the base of S2a is concordant on the folded Urgonian whereas sequences (S2b?)-S3 exhibit growth-strata geometries inferring a westward thickening of sedimentary deposits (Fig. 14C, D), which can be attributed to an anticline formation in the hangingwall of FZ3. Without further stratigraphic constraints, however, we can only speculate that FZ3 is active at least since deposition of S3 (~16.2 Ma, Fig. 17). On the eastern edge of the Bas-Dauphiné basin, at the footwall of FZ4, the presence of growth-strata geometries between sequences S4 and S5 (Fig. 14A), as well as the occurrence of a S4 depocenter between FZ4 and FZ5 (Fig. 15), implies a flexural subsidence at the footwall of the active FZ4, therefore constraining the activity of FZ4 between the onset of S4 and the end of S5, so between ~15.0 and ~12.0 Ma (Fig. 4A, B; Fig. 17). This is also in agreement with the underlying S2 and S3 deposits, and the overlying S6 deposits, which all appear isopachous (Fig. 14, Fig. 15), and thus suggests the absence of tectonic activity during deposition of these sequences. At the footwall of FZ5, the presence of a probable S5a-S5b depocenter (Fig. 15C) may indicate that the depocenter of S5 (~14.0 to ~12.0 Ma) was mainly controlled by the FZ5 activity, and would thus further illustrate a progressive westward migration of the successive depocenters (Fig. 17). This hypotheis is consistent with the general trend observed since the base of S2a at the zone A, 18.05 +/- 0.3 Ma ago, which shows the successive activation of the fault zones from east to west (SAL fault, GF fault, FZ2, FZ3, FZ4 and finally FZ5). Thus, the thrusts that root in the Belledonne thrust were activated in-sequence (Fig. 17). The S6 deposits constant thickness in the Bas-Dauphiné basin (Fig. 15), and the S6 sealing the underlying growth strata (Fig. 14A), suggest an end of the Phase 2 at ~12.0 Ma (top S5), in the subalpine massifs.

### 5.1.4 The Jura thrust activity synchroneous to a regional scale tectonic uplift? (Phase 3)

To the east of the La Bresse basin, the S6 and S7 marine deposits (~12 to ~10Ma, Fig. A9) are thrusted by the Jura thrust (at the BY-101 locality, Fig. 1B) indicating that the onset of the Jura thrust is either synchronous, or posterior, to sequences S6-S7. Due to the occurrence of abnormally thick Tortonian S7 and S8 continental deposits (~10 to ~8.2 Ma) at the footwall of the Jura thrust suggesting a tectonic-controlled depocenter, we suggest that the Jura frontal thrust (J, Fig. 1B) is most likely active during S8 (~ 9.5 Ma; Fig. 4A), perhaps as soon as the S7 continental regressive deposits (~ 10Ma).

This hypothesis is further supported by earlier chronostratigraphical studies based on mammals, suggesting that the activity of the Jura frontal thrust (J, Fig. 1B) started between the Serravalien (~11 Ma) and the Pliocene (Bolliger et al., 1993; Steininger et al., 1996; Kälin, 1997). U-Pb ages of syntectonic calcite mineralizations sampled from the Buron thrust (Bur, Fig. 1B) give an age of 10.6 ± 0.5 Ma (Smeraglia et al., 2021). Further north, the same U/Pb ages of syntectonic calcite mineralizations are 11.4 ± 1.1 Ma (Montlebon thrust), 9.7 ± 1.4 Ma, and 7.5 ± 1.1 Ma (Arguel thrust, northward prolongation of J, Fig. 1B) from SE to NW suggesting an in-sequence thrust propagation within the Jura fold-and-thrust belt (Smeraglia et al., 2021). Paleontological evidences combined with tectonic evidences indicate a termination of Jura thin-skinned foreland tectonics between 9 and 4 Ma (Becker, 2000).



In the Bresse, the western and southern parts of the Bas-Dauphiné basin and the Crest basin (respectively, I, J, K, L zones, Fig. 3), where the influence of the Jura thrust cannot be invoked, the Miocene final sea retreat is recorded during deposition of S7 (Fig. A9, 10, 11, 12). The absence of marine deposits during S8 is unexpected, as the eustatic transgression that
occurred during S8 was characterized by a greater elevation of the sea level (+40 m) than for S7 (+5-10 m, Miller et al., 2005, Fig. 4B). So, the Miocene sea-retreat during S8 (or sooner, during the S7 regressive deposits) was probably induced by a basin-scale factor.

In the north-eastern part of the Bas-Dauphiné basin (F, H zones, Fig. 5), the seismic profile across the Chartreuse front (Fig. 14) suggest the absence of a post-S5 (~14 to 12 Ma) thrust activity. There, the toppest marine deposits (S6 regressive
deposits, Fig. A6, 8) outcrop today at elevation of ~350 m.a.s.l. (Fig. 14B), which suggests a post-S6 minimum uplift of ca. 350 m. We thus propose that this uplift (or part of it) promoted the rapid regression of the Miocene Sea recorded in S7 in the western and southern parts of the Bas-Dauphiné basin, which approximately coincides with the onset of the Jura thrust. This adds new evidences to Deville et al. (1994)'s observations postulating for a post-Langhian uplift (linked with an isostatic rebound) and the activation of out-of-sequence thrust systems in an internal position of the subalpine massifs. Interestingly,
in the North Alpine foreland basin, Mock et al. (2019, 2020) documented a similar pattern based on low-temperature apatite (U-Th-Sm)/He thermochronometry. Between 12 Ma and 5 Ma, Molasse deposits were affected by thrusting along the ECMs borders and the entire front of the Central Alps, coeval with the main deformation phase of the adjacent Jura massif and the late exhumation of the ECMs. Hundreds of kilometers southwestwards, our results therefore support that this regional uplift probably corresponds to a tectonic event at the scale of the Alps, which may be caused by lithospheric processes (Mock et
al., 2019, 2020).

**5.2 Sequence of shortening of the Southern Jura and Subalpine massifs**

To further quantify the amount and history of shortening, cross-sections B and C (respectively, northern and southern Chartreuse, Fig. 11B, C) were balanced and restored to the initial geometry before the alpine shortening. Such reconstruction takes into account the relative timing of the various faults and yields a reconstruction of the deformation through time as well
as an initial state. The cross sections have been drawn perpendicular to the main thrusts and fold axis (Fig. 1B), N118° for cross-section B and N114° for cross-section C. Total horizontal shortening along the cross-section B is 22.24 km (33.9 %) and 13.72 km (22.5 %) along the cross-section C (Fig. 18). These values are somehow approximate since they result from the hypothesis of a planar strain along a single and uniform direction of deformation for each section. They are in agreement with the previous finding of an increase in the shortening toward the north (Menard and Thouvenot, 1987; Sinclair, 1997;
Philippe et al., 1998) that is interpreted as reflecting a larger amount of convergence (Bellahsen et al., 2014).

The sequential reconstruction allows to calculate the shortening related to each fault and fold and our chronological constrains on the thrusts bounds the quantification of some deformation rates. The amount of horizontal shortening for the whole Phase 1 (i.e. FZ1 horizontal shortening), is 14.1 km along the cross-section B and 5.8 km along the cross-section C, (Fig. 18A, B, Table S6). Note that the backthrusts visible in the east of section C (bk1 and bk2, Fig. 11C; Fig. 18B) are





excluded from that calculation as they are only seen along that section and considered as related to another tectonic phase. The amount of horizontal shortening for the whole Phase 2 (i.e., GF thrust, FZ2, FZ3 and FZ4 horizontal shortening) is of 6.7 km in cross-section B and 6.3 km in cross-section C (Fig. 18A, B, Table S6). In the Chartreuse massif, the SAL fault wasn't described (Fig. 1B), implying that Phase 2 lasted between 17.35 +/- 0.15 Ma (onset of the GF fault) and ~12 Ma, thus for ~5.3 Ma. This implies a migration of the deformation toward the external parts (west) at a rate of ~2.9 km Ma$^{-1}$ along

sections B and C. This also corresponds to average shortening rates of 1.26 km Ma$^{-1}$ for cross-section B and 1.19 km Ma$^{-1}$ for cross-section C.

**Figure 18: Balanced cross-sections throughout the Chartreuse massif using the Move software. To facilitate the restauration, we assume that the ZF2 to ZF4 are rooted below a single Belledonne basal thrust. (A) Cross-section B (Fig. 11B). (B) Cross-section C (Fig. 11C). See Fig. 1B for locations.**



## 6. Conclusions

This study focused on the Miocene molasse of the western Alpine foreland basin (Subalpine massifs, southern Jura, the Bas-Dauphiné, the la Bresse and the Crest basins). The combination of chemiostratigraphy, biostratigraphy and magnetostratigraphy applied on sedimentological sections and well logs from the Miocene succession across the entire area allowed the establishment of a new chronostratigraphic framework. The Miocene is characterized by 11 depositional sequences dated between the late Aquitanian and the Tortonian. The spatial distribution of these sequences defines 12 depositional zones grouped into 4 domains. These domains are bounded by the main thrust fault zones, therefore implying a strong interplay between active tectonic and sedimentation during the Miocene, and lead us to propose three main tectonic phases (Fig. 17).

(i)     Phase 1 took place between the early Oligocene (29 +/- 2 Ma) and the early Miocene (~17.8 Ma). During this phase, the southern Jura synclines were affected by flexural subsidence induced by the alpine tectonic front to the east. The maximum depositional thickness recorded to the east of the Rumilly syncline, as well as the absence of early Miocene marine deposits to the east of the FZ1 most probably indicate that the tectonic activity was along FZ1 rooted east of the Belledonne external range.

(ii)    Phase 2 took place between the late Burdigalian (18.05 +/- 0.25 Ma) and the Serravalian (~12 Ma). This phase corresponds to the activation of the Belledonne basal thrust fault, rooted below the external Belledonne massif, and to the propagation at an average rate of ~2.9 km Ma$^{-1}$ of the deformation toward the west within the subalpine ranges (successively SAL fault, GF fault, FZ2, FZ3, FZ4 and FZ5 thrusts). In the Chartreuse massif, the shortening rate related to that phase averages ~1.2 km/Ma. At 18.05 +/- 0.25 Ma, the development of a depocenter at the footwall of the SAL thrust along with the occurrence of seismites (of regional extent) together suggest the activation of the SAL fault. Subsequently, at 17.35 +/-0.15 Ma, an angular unconformity characterized by westward-directed onlaps is recorded. This angular unconformity discloses a brutal westward migration of the depocenter at the footwall of GF fault and thereby points to its activation. At ~16.2 Ma, growth strata relationships in S3 (+S2b?) in the hangingwall of FZ3, combined with the occurrence of a S3 depocenter in the footwall of FZ2 argue for the quasi-synchronous activation of FZ2 and FZ3 thrusts. Subsequently, at ~15.0 Ma, S4 growth strata geometries and the development of a S4 depocenter in the footwall of the FZ4 document the onset of FZ4. Finally, at ~14.0 Ma, the S5 depocenter occurs in the footwall of FZ5, and probably dates the activation of the FZ5 thrust. The end of the Phase 2 took place at ~12.0 Ma, as the S6 deposits have a constant thickness in the Bas-Dauphiné basin, and S6 seals the underlying growth strata (S2 to S5).

(iii)   The deformation lasted longer in the Jura. Thick accumulation of continental S7 and S8 deposits at the footwall of the Jura frontal thrust is coeval with thrusting of Mesozoic series atop S6 and S7 marine deposits (~12 to ~10 Ma) indicating that the Jura front initiated during the Tortonian (~10 Ma, phase 3). Synchronously, to the



south in the Bas-Dauphiné zones and the Crest basin where no thrust activity was recorded post-S5 (~12 Ma), the third phase is characterized by an uplift, as the absence of marine deposits during S8 contrasts with a significant eustatic flooding at that time (Miller et al., 2005). Such a regional uplift coeval to the onset of the Jura frontal thrust (J) coincides with an alpine-scale tectonic event, since thrust-driven exhumation have been concurrently reported in the north Alpine foreland basin between 12 and 4 Ma (Mock et al., 2019, 2020).









**Figure A1: Synthetic section of the Miocene of the Rumilly-Chambéry synclines (Zone A). See Fig. 3 for location of the sedimentological sections and Fig. A13 for legend. For more details, see section 6.3.1. and appendix 17 in Kalifi (2020) and references therein.**


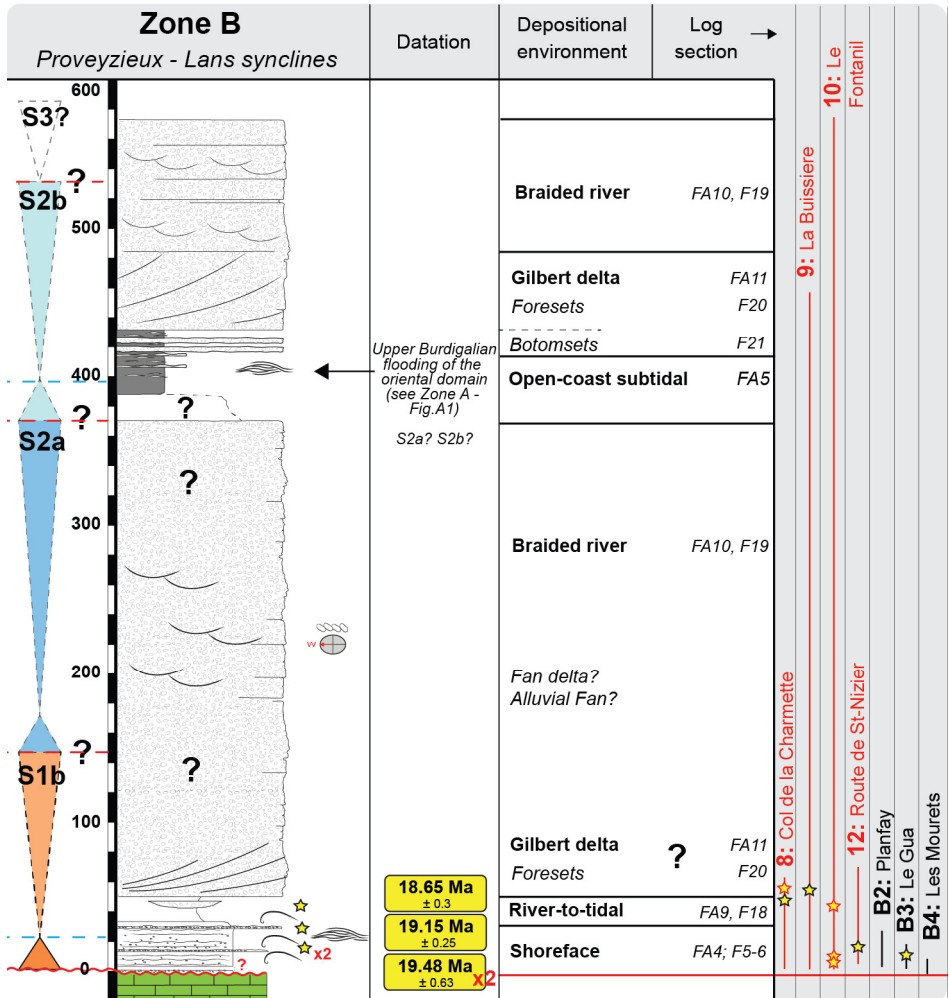

**Figure A2: Synthetic section of the Miocene of the Proveyzieux-Lans synclines (Zone B). See Fig. 3 for location of the sedimentological sections and Fig. A13 for legend. For more details, see section 6.3.2. and appendix 26 in Kalifi (2020) and references therein.**








Figure A3: Synthetic section of the Miocene of the Novalaise syncline (Zone C). See Fig. 3 for location of the sedimentological sections and Fig. A13 for legend. For more details, see section 6.3.3. and appendix 30 in Kalifi (2020) and references therein.



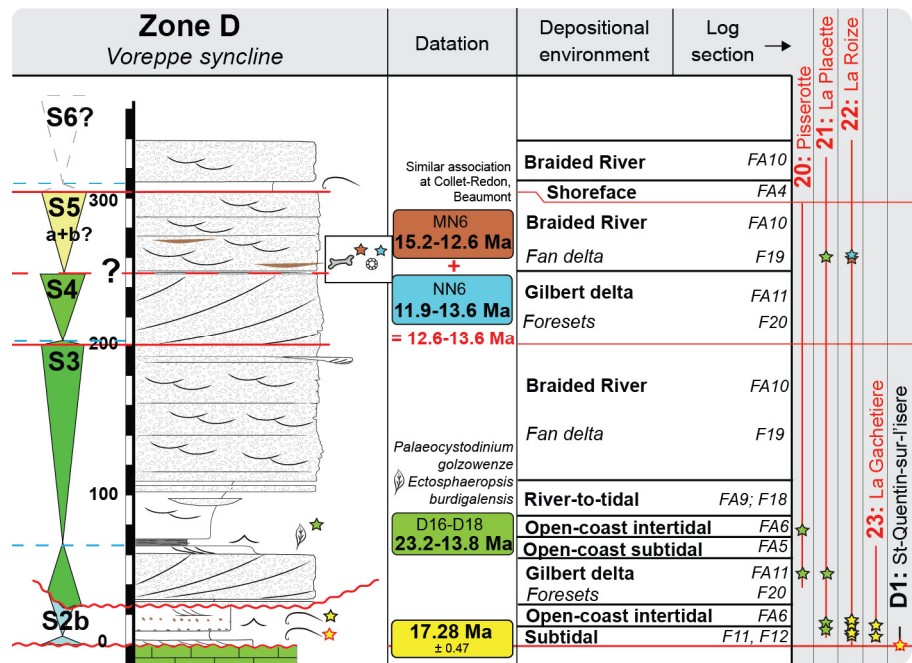


**Figure A4: Synthetic section of the Miocene of the Voreppe syncline (Zone D). See Fig. 3 for location of the sedimentological sections and Fig. A13 for legend. For more details, see section 6.3.4. and appendix 39 in Kalifi (2020) and references therein.**

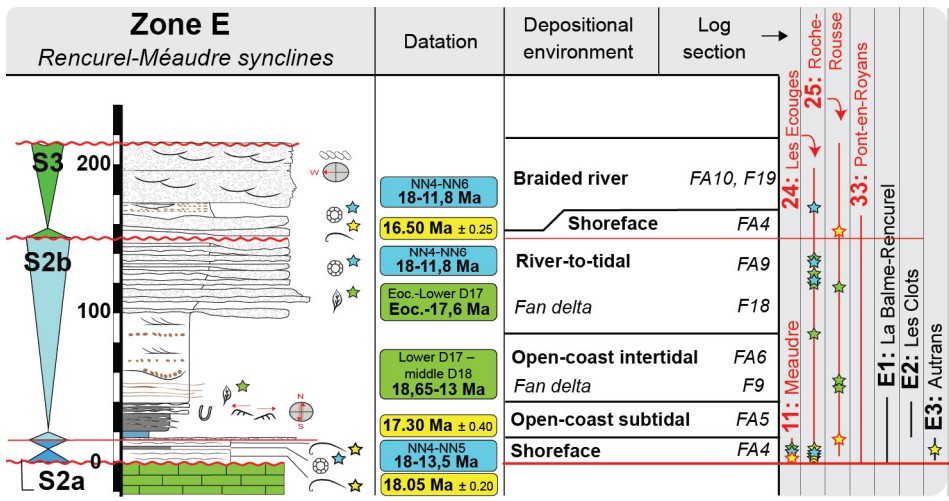

**Figure A5: Synthetic section of the Miocene of the Rencurel-Méaudre synclines (Zone E). See Fig. 3 for location of the sedimentological sections and Fig. A13 for legend. For more details, see section 6.3.5. and appendix 44 in Kalifi (2020) and references therein.**





Figure A6: Synthetic section of the Miocene of the Bièvres region (Zone F). See Fig. 3 for location of the sedimentological sections and Fig. A13 for legend. For more details, see section 6.3.6. and appendix 48 in Kalifi (2020) and references therein.



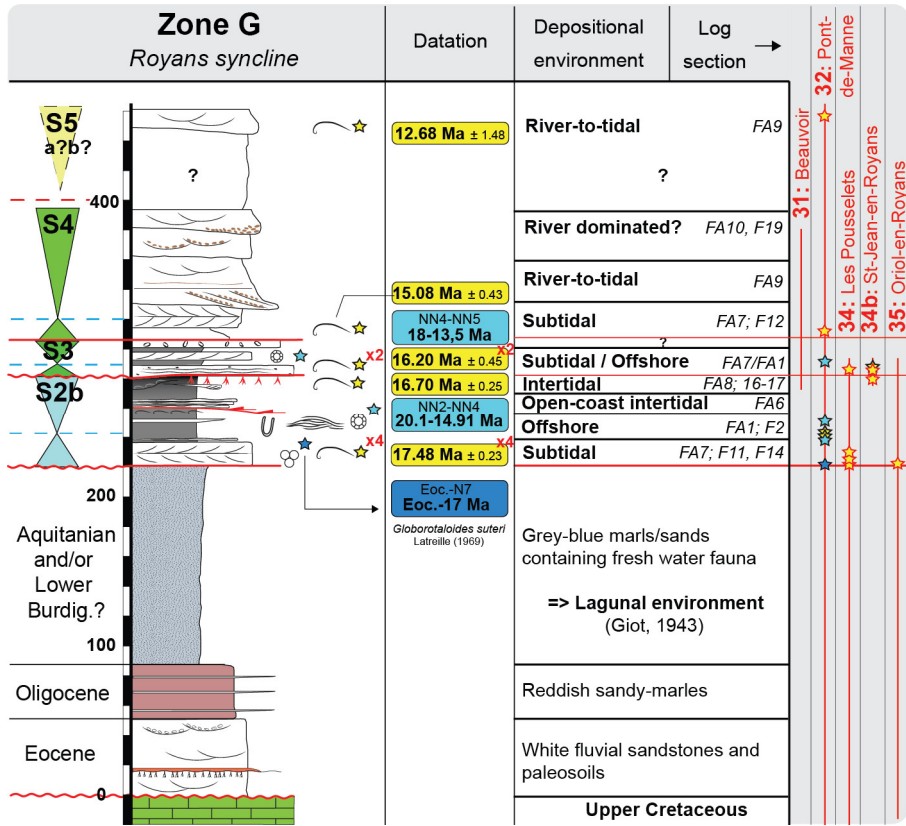

**Figure A7: Synthetic section of the Miocene of the Royans syncline (Zone G). See Fig. 3 for location of the sedimentological sections and Fig. A13 for legend. For more details, see section 6.3.7. and appendix 54 in Kalifi (2020) and references therein.**





Figure A8: Synthetic section of the Miocene of the La-tour-du-Pin area (Zone H). See Fig. 3 for location of the sedimentological sections and Fig. A13 for legend. For more details, see section 6.4.1. and appendix 59 in Kalifi (2020) and references therein.


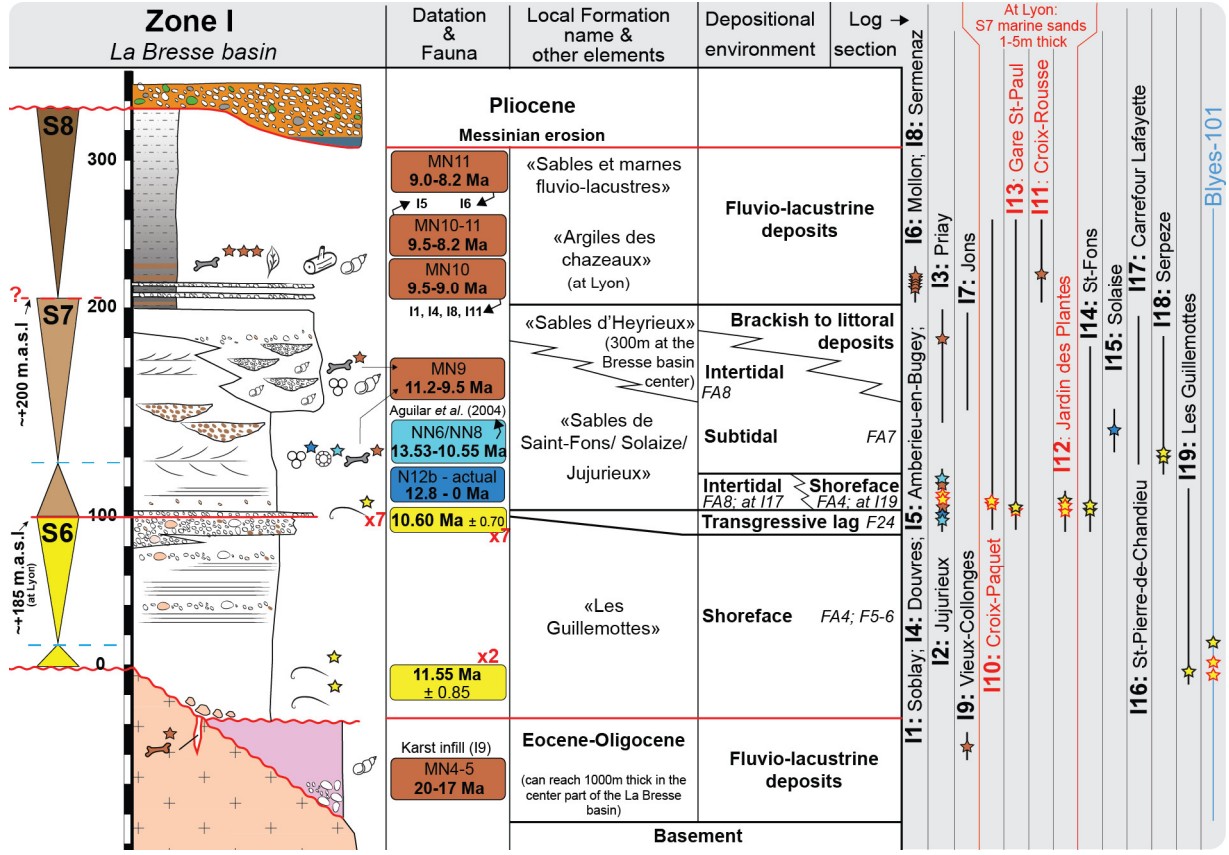

**Figure A9: Synthetic section of the Miocene of the La Bresse basin (Zone I). See Fig. 3 for location of the sedimentological sections and Fig. A13 for legend. For more details, see section 6.4.2. and appendix 62 in Kalifi (2020) and references therein.**







Figure A10: Synthetic section of the Miocene of the Bonnevaux plateau (Zone J). See Fig. 3 for location of the sedimentological sections and Fig. A13 for legend. For more details, see section 6.4.3. and appendix 65 in Kalifi (2020) and references therein.






**Figure A11: Synthetic section of the Miocene of the Chambaran plateau (Zone K). See Fig. 3 for location of the sedimentological sections and Fig. A13 for legend. For more details, see section 6.4.4. and appendix 69 in Kalifi (2020) and references therein.**








**Figure A12: Synthetic section of the Miocene of the Crest basin (Zone L). See Fig. 3 for location of the sedimentological sections and Fig. A13 for legend. For more details, see section 6.4.5. and appendix 82 in Kalifi (2020) and references therein.**





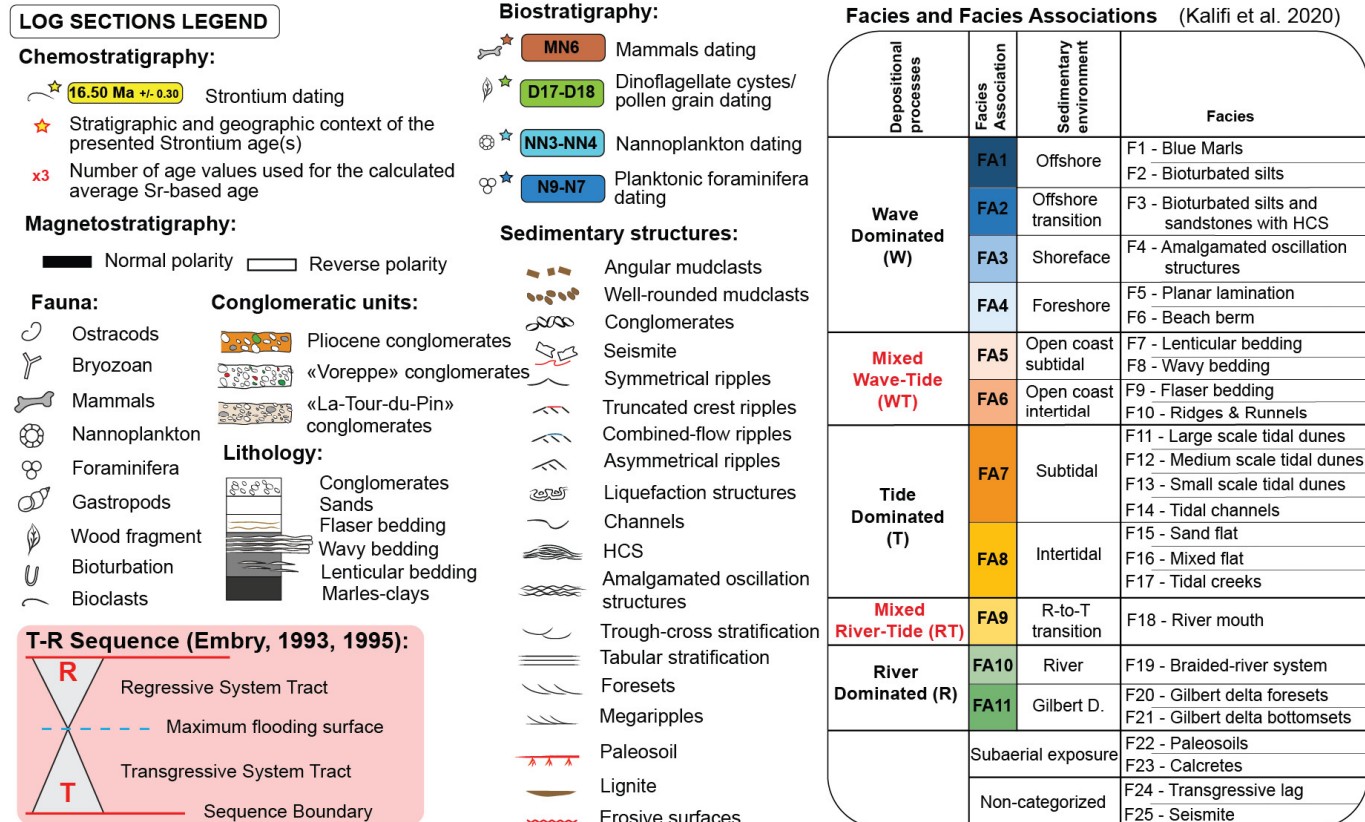

**Figure A13: Legend for synthetic sections of the 12 palaeogeographical zones of the Miocene basin of the subalpine massifs, southern Jura and adjacent basins (Bas-Dauphiné, La Bresse, Crest) provided in appendix A.**





# Appendix B: The FZ1 along the western edge of the Belledonne massif

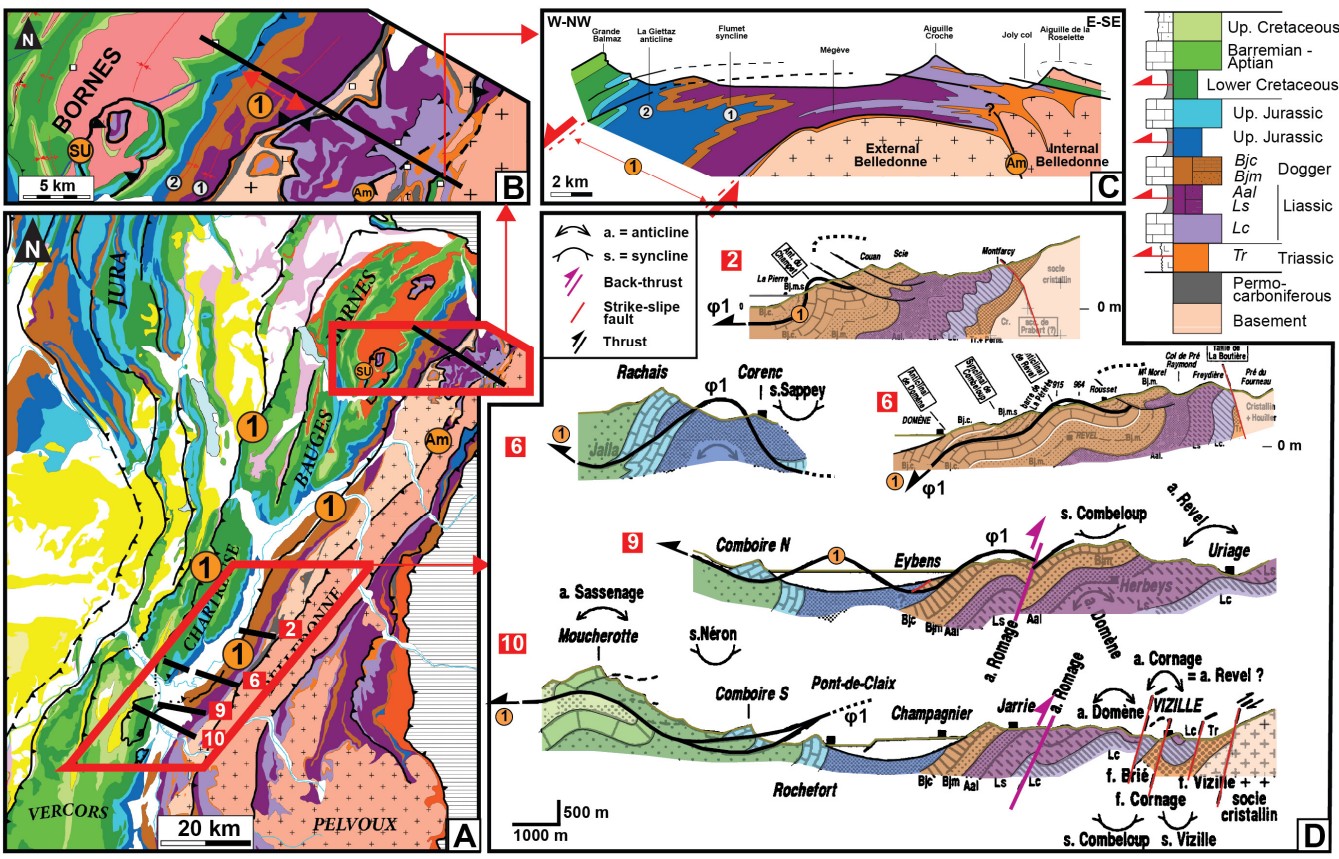

Figure B1: The FZ1 along the western edge of the Belledonne massif (A) Localization of the cross-sections; (B) Close-up view of
the FZ1 between the eastern border of the Bauges and Belledonne massifs. Localization of the ESE-WNW cross-section of Gidon
(2004) presented in (C). SU= Sulens klippen. Am= Internal Belledonne thrust. (D) The FZ1 along the Gresivaudan hills adapted
from Barféty and Gidon (1996).




*Data availability*. The research data are included in this paper and in the Supplement and can be freely accessed.

*Supplement*. The supplementary materials related to this manuscript are available in the attached .pdf file.

*Authors contribution*. This study was performed by AK in the framework of his Ph.D, co-supervised by TOTAL (CSTJF, Pau, France) and the LGL-TPE (University Lyon 1). JLR conceptualized the original research topic of this study, with the support of VS, in the EXPLO/EP/GTS/ISS entity of TOTAL. Fieldwork and interpretations related to the

chronostratigraphical study were performed by AK under the supervision of PS, BP, JLR, and by BH under the supervision of JLR and AK, for the northern part of the Bas-Dauphiné basin. Fieldwork and interpretations related to the structural study were performed by AK under the supervision of PHL, and the helpful support of VS. Restauration of the structural cross-sections were conducted by KL supervised by PHL and AK. The sampling and the sample preparation for the chemostratigraphical study was achieved by AK, and the analysis and interpretations were conducted by AG. The sampling

for the paleomagnetic study was carried out by FD, PS, RG and AK. AK prepared the samples and the interpretations were conducted by FD. The sampling for the biostratigraphical study was conducted by AK, BH and RG, and the interpretations were performed by FQ for the planktonic foraminifera, by DM for the dinoflagellate cysts and by FR for the calcareous nannofossils. AK writed the paper, with major contributions of PH, PS and BP, and corrections from all other all co-authors.

*Competing interests*. The authors declare that no competing interest are present.

*Acknowledgments*. The authors would like to thank TOTAL for the financement of this PhD study. We would like to thank Andrea-Lopez Vega, Thomas Pichancourt, Hawoly Bass, Astrid Jonet, Antoine Mercier, Alessandro Menini, Pierre and Thomas Courrier, Daniel Fournier, Ludovic Mocochain, Sidonie Revillon, Hugues Fenies, Gilles Escarguel, Loïc Costeur,

Jonathan Pelletier, Edouard Le-Garzic, Xavier Du-Bernard, Jean-Pierre Girard and Francois Lafont for their precious help in the field and/or for fruitful discussions.



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
