# Peer review of "Chronology of thrust propagation from an updated tectonosedimentary framework of the Miocene molasse (western Alps)"

_Solid Earth, 2021_

## Referee Comment (RC1)

Review of the manuscript **"Chronology of thrust propagation from an updated tectono-sedimentary framework of the Miocene molasse (Western Alps)",** Kalifi et al., submitted to Solid Earth, july 2021
Thierry Dumont, CNRS, ISTerre, Université Grenoble Alpes

**Overview**

The topics of this manuscript is of major interest, since the Tertiary sedimentation allows to constrain the orogenic propagation in the Alpine foreland. The core of the manuscript consists of integration of new geochronological data (biostratigraphy, chemostratigraphy, magnetostratigraphy) with the existing database in a sequence statigraphic framework, along with a synthesis of available well-log and seismic profiles. Besides this, the authors provide a reappraised structural framework based on existing maps, subsurface information from key seismic profiles, and field overview of some key areas. This reappraisal also benefits from some previously published field sections or even unpublished elements from the geol-alp.com website. These sedimentary and structural synthesis are correlated to propose a dynamic tectono-sedimentary and paleogeographic framework of the forward propagation of Alpine orogeny since Oligocene times.
This work is clearly worth of publication, provided it takes into consideration the comments listed below. The most reliable and solid input is the chronostratigraphical synthesis, and the important information brought by field sections and wells/profiles analysis. I have more reservations about the structural synthesis, which lacks kinematic data about brittle deformation (thrusts, faults), folds analysis, ductile deformation (Bornes) and transport directions, which also lacks 3D maps analysis, and which attempts anyway to conclude about stress evolution and chronology of thrusting. Thus, some conclusions appear overinterpreted, such as the complete allochtony of the northern Subalpine massifs, or the attribution of an Oligocene age to the earliest identified thrust (the paper is furthermore devoted to Miocene). From geodynamic point of view, the demonstration and time-space quantification of the forward propagation of the Alpine front since early-middle Miocene is convincing, although the geodynamic and structural inheritance and specificity of the Oligocene phase, along with older inherited structures (Hercynian trends), could have been better considered.

**General comments**

*Organisation* of the manuscript is correct but the "Geological setting" and the "overall structure.." § partly overlap concerning the description of main thrusts, their description in §2 could be simplified as both are describing fig. 1B; lines 115 and following partly duplicate § 4.3 (i.e. "the southern prolongation of this fault is contentious" found line 120 and line 348).
The structural descriptions of §4.3 are long and tedious, even the 4.3.6 "summary" trying to justify the options choosen in the cross-sections. Could be better organised and shortened.

*Stratigraphy, sedimentology*
This is of course the main input of the paper, thanks to field sections and synthesis of borehole data and seismic profiles. Synthetic sections provided in appendix are original and essential. The chronostatigraphic integration of different methods seems solid, and provides an essential framework to analyse the tectonosedimentary features.

*Structures and deformation*
Despite the structural study refers to "new field data", synthetic presentation of these new data is lacking (i.e. line 389 "210 stations.." without location location map nor data synthesis). Kinematics and structural chronology are tentatively deduced from maps analysis (fold and thrust trends), which is not suitable for proper identification of stress directions and discrimination of  deformation phases. There is confusion between finite deformation (folds, thrusts) and "shortening phase", suggesting that paleostress can be inferred from the present structural trends, which is abusive. Thrusts and folds are oblique (eastern thrust in Chartreuse is western in Bauges) and sometimes curved. Such variations could be explained by structural inheritance beneath the foreland, with the possible influence of basement inherited structures oblique to Alpine stress, however this would require a specific microtectonic analysis is to determine paleostress, which obviously is not the main aim of this paper. Nevertheless, such analysis will now be facilitated by the improved chronostratigraphic framework provided by it.
Some more specific comments about FZ1:
The connection of thrust 1 both to the western Bauges and to the eastern Bauges implies that the Bauges and Bornes are regarded as an allochtonous nappe. This extreme opinion should be documented by structural observations in the Eastern Bauges massif, which are lacking. Although deformed, the Jurassic cover near Ugine is not detached but only affected by distributed shear. The autors use the data of Gidon's website (i.e. fig. B1C = "unpublished" http://www.geol-alp.com/h_mt_blanc/_schemas/coupe_Aravis_mtBlanc_4.gif) but Gidon himself does not consider that the sedimentary cover of the Bornes massif is detached, although it looks sheared and deformed. The authors should provide additionnal structural data from their own to support the detachment interpretation, or give a better consideration to ductile deformation.

Thrust 1 shows a strong lateral increase in amplitude from S to N. It seems that accomodation is increasing northwards during the deposition of sequences S2 and S3, maybe the correlation between both should be underlined.

The onset of activity of FZ1 ("phase 1) is proposed as early Oligocene (5.1.1, fig. 16). I do not understand the argument for this? 1) eastward thickening of the Oligocene sequence in profiles 10B may be a flexural response related to more internal structures than FZ1, 2) Oligocene series are transported in the hangingwall of FZ1 (Bauges), 3) in the southern termination of FZ1 around Grenoble, there is absolutely no evidence for Oligocene activity, the thrust is overlying Aquitanian/Burdigalian, 4) early Oligocene corresponds to the activation of the Penninic thrust, further deformed by FZ1. This opinion must be better discussed and justified, and the "P1" bar in dashed line for Oligocene.

Other ZF:

The connexion from the Subalpine front to the Jura debatable:

The S-N continuity of thrust zones 2-3 remain a matter fo debate even with the help of seismic profiles (i.e. FZ2 discussion p. 22)

The offset of FZ4 is much greater to the S (Royans) and tends to lower in front of N Vercors and Chartreuse (fig.9), this makes its northern cotinuity towards Jura questionnable.

The extent of FZ5 to the N is not well constrained (line 522)

Moreover, the connexion of FZ2-FZ3 towards Jura + the interpretative roots of FZ2-FZ3 beneath Belledonne would imply that the Jura thrusts are rooted beneath N Belledonne, that is 150km further SE..

More generally, the boundaries of the structural domains appear to be defined and chosen "à priori" by map synthesis (fig. 1B, §2), so that the final identification of thrust activity is partly a circular argument.

*Geodynamics*

Oligocene is a very specific period from geodynamic point of view (west European rift system, Ligurian sea rifting) and the study area is located at the hinge between the Alpine orogen and the rift system. The inheritance of Oligocene structures and paleogeography could be better introduced. More specifically, I think that the "forebulge" interpretation put forward in section 5 and fig. 17 is, to me, somehow model-driven. The study area is large, and it would be worth to distinguish the flexural foreland situation in front of the Swiss molasse basin to the NE, from the Rhone valley rift to the SW. While a forbulge uplift seems reasonable in the former case (section A), the Royans Oligocene paleorelief is more probably related with the large-scale half-graben structure nicely illustrated in fig. 9D, whose Miocene reactivation of the eastern part localized FZ4-FZ5. I would not identify this structure as a forebulge.

**Detail points**

49-51 N-NW directed from Eocene to earliest Oligocene, driven by Adria-Europe convergence, then W directed motion driven by extrusion of the internal W Alps, leading to the radially propagating arc.

"..Vercors and Chartreuse.."

"..NNW-SSE trending Miocene thrusts.."

"..date from the late Burdigalian.."

"Further south, .."

"..well-logs and field sections."

"as a response to the western propagation of the Alpine orogen during the Miocene."

also quote De Graciansky, P.C. de, Roberts D.G., Tricart P. (2011), The Western Alps, from rift to passive margin to orogenic belt, an intergated overview. Developments in Earth Surface Processes, 14, Elsevier, 398p., ISBN 9780444537249

"..the westward progressive migration.."?

fluvial deposits?

"..folds and thrusts affecting the sedimentary cover, trending NNW-SSE.." some folds are ~NS? "last WNW-ESE shortening phase" is overinterpreted, paleostress cannot be assumed to be perpendicular to present fold trends, nor can be determined along balanced cross sections whose orientation was choosen à priori.

please locate (from N to S of what?)

a more recent ref. about the Moucherotte thrust is still missing (Donzeau, Gamond & Mugnier, CRAS 1993 vol 317 p. 1675)

split the sentence after Gidon, 1964

"..the Jura fold belt is progressively widening northwards."

"..anticlines that develop in the hangingwall of blind thrust.."

also quote Lickorish et al. (2002) GSA Bulletin 114, 9, p. 1089

"..from 35 well-log and outcrop sections.."

resolution?

"..in the footwall of the Penninic thrust.."

"..thrust over.."

not sure that "a saddle of the folds" is correct..

"new field observation" should require structural measurements, only landscape views are presented.

"suggest" incorrect does it mean that the Sassenage anticline is partly older than Miocene? to be demonstrated, fig. 6B does not show anything

"On each side.."

how can you determine fold trends so precisely without structural data? not visible in fig. 7a a better argument than the map trace would be a view of 3D geological map this is already proposed by Gidon 1995, fig. 1, must be quoted is this left-lateral displacement necessary considering the dip of the Neron-Moucherotte thrust and the topography? 3D view would be useful what means "..at 210 stations"? where are the location/data? should be removed if it is not presented. I do not find this detailed map of the Moucherotte thrust. Would require at least 3D map view for discussion

"..thrust above the Miocene on top of.."

this thrust at the base of Comboire cliff is not the main Moucherotte thrust because the footwall rocks are Berriasian, vhereas the footwall of the main thrust should be younger (Valanginian, see Gidon, geol-alp website, page Comboire, and quote it). The argument is difficult to understand from your text.

".. overlain by.."

fig. 6C? "..the lower Cretaceous strata are steeply dipping eastwards"

text description very difficult to follow

"..in the footwall of the thrust."

"extend"

"..but dips locally to the west, strongly suggests that.."

in contradiction with the trace of the "Corenc tectonic window"..

no, "External Belledonne" is basement, here this is cover

424-425 the displacement along the "Corenc-Jalla" thrust seems much lower compared to Neron

I do not understand this sentence, please clarify there seem to be confusion between a "thrust" and a shear zone, what is the thickness of the folded section?

Bellahsen et al do not figurate a complete detachment across the Bauges massif once again structural data should be synthetised in stereograms, not as individual numeric values in the text.

may be this can be interpreted as southward decrease in thrust displacement as for FZ1?

veers? sentence unclear this is the classical view from the Grenoble 1/50000 sheet and Gidon papers, see also Dumont & SPIA, 2020, Geol Soc London Spec Publ 486 doi:10.1144/SP486-2019-92

"Its interpretation suggests that.." Complicated discussion, it appears that the S-N continuity of the individual thrust zones is interpretative and debatable, even with the help of seismic profiles.

not found the OU thrust in fig. 9B, inherited from Deville 2021? check consistency: Outherans/Outeran, SA anticline not on fig.10, FZ1 text is 1 on fig.10.

why "west-dipping GF"? seems to be E-dipping everywhere

I do not understand, the potential connection is not visible on the profiles of fig.10

"..by steep west-dipping.."

"..offset is estimated to.."

508-511 major thrust .. (FZ3) / the main fault (3) : be consistent in the text description offset along 4 + 4a on line 82... looks greater than 650m ($\sim$1,5km?)

526-530 this presentation is a bit oversimplified, both thrust traces and fold trends vary between NS and NE-SW, and even NW-SE following the interpretations about the FZ continuity in Jura.

further north what? BZN1 quite impossible to locate on fig.1, too small. What is the importance of this triple occurrence of Js? ("duplicated three times" means that Js occurs 6 times?)

"an uplifted compartment" where?

quote fig. 18 or remove?

"Klippe" is just a result of erosion, it does not have any implication in terms of displacement or location of "roots"

.. or simply that Subalpine and Jura thrusts cannot be individually connected?

also quote Lickorish et al., 2006; Deville 2021.

veer?

566-572 this discussion lacks kinematic data, in the absence of which it is not possible to decipher the relative chronology between thrusts and transfer/strike-slip faults. Similarly, the orientation of compression cannot be constrained from maps only. Do the southern Jura thrusts imply a NE-SW compression?

"In the footwall of FZ1.."

"..in the footwall.."

".. the deposition of Oligo-Aquitanian sediments"

"..both Oligocene and early Miocene sediments.."

"are proposed using the.."

".. while the regressive systems tract.."

(and elsewhere) "Therefore, the sequences S6 and S7..."

"..is either synchronous or younger than the deposition of sequences .."

and §5.1.1 there is no direct evidence of the activity of FZ1 as soon as early Oligocene, the Oligocene sedimentary sequence is compatible with a flexural foreland but, as said line 763, the eastern boundary of this basin is unknown. Moreover, Oligocene is transported in the Bauges massif, suggesting that the orogenic front was located further east. This option should be presented as an hypothesis and the phase 1 (P1) on fig. 16 put in dashed line.

what means « concordant on the folded Urgonian », is Miocene folded with Urgonian ?

the wavelength of flexural subsidence is several hundred km, consistent with the lithospheric thickness, this process can hardly be involved for a depocenter in the footwall of an individual thrust.

hypothesis

"..the thrusts that are interpreted to root beneath the Belledonne massif.."

uncorrect sentence (and the S6 sealing…)

"end of phase 2" is time, "in the subalpine massifs" is location, unclear

"..are overthrust by the Jura units.."

posterior younger than..

tectonically-controlled..

"..even perhaps coeval with the deposition of S7 continental sediments"

"..suggests that thrust activity stopped after the deposition of S5. There, the uppermost marine.."

the link between the arguments presented since line 828 and "activation of out-of-sequence thrusts in an internal position" is unclear. Moreover, such structures should produce subsidence in their footwall, not uplift; and isostatic rebound is mentioned line 833. So please clarify your interpretation of a "tectonic event" involving "lithospheric processes", it is now clear that the recent uplift of the W Alps is not directly linked with plates convergence, and is probably not "tectonic".

backthrusts are compatible with thrusting, they do not need another tectonic phase?

the onset of phase 1 as soon as early Oligocene is not well constrained, must be given as hypothetical.

875-880 there is an overlap between phase 1 and 2? (17,8Ma+/-?? - 18,05+/-0,25Ma), better use Early Burdigalian line 875?

the "Belledonne basal thrust fault" is actually unknown, no outcrop, no seismic, no borehole. Must be presented as an interpretation.

"..recorded after the deposition of S5"

901-903 I do not understand which tectonic structure could be involved to explain this uplift by tectonic processes in Bas Dauphiné? there is no evidence of shortening, and the situation is different from the N Alpine foreland.

**Figures**

Figures are generally of good quality, and contain a lot of information, but will be hard to read on a paper copy.

Figs 4 & 5 dense but clear, this is the core of the data

Figure 6: A and B are not very clear because of backlight, a 3d view of the geological map draped over DEM would perhaps give a better overview. Picture H can be processed to get better colours

Fig. 7: Why are the cross-sections named both with numbers (1-6) and letters (B-G)? confusing. I do not understand the trace of the east part of FZ1 outcropping both W of Corenc and E of Grenoble, not compatible with cross sections 1 & 2.

Fig. 9: are the interpretation based on Deville 2021 or were the lines completely reinterpreted? must be clarified

Fig. 11: I remain reluctant considering the FZ1 detachement over Belledonne in section A, and I totally disagree with its trace above Belledonne in section D: the normal fault offset is Jurassic as demonstrated in the 80s (La Mure fault) as the Liassic series have very different thickness on both sides (Lemoine et al., 1986). Must be changed.

Fig. 13: sections identified using both n° and letters (3 = A, 4 = B etc.), keep only n°?

Fig. 15 please refer to the location of transects on a map

Fig. 17: the activation of FZ1 to the east is not consistent with the occurrence of Oligocene deposition in the futuer Bauges massif, that is in the hangingwall of the thrust. The activation of the Belledonne thrust as soon as ~20Ma is in contradiction with the exhumation ages which are younger.

Fig. 18: (A) = cross-section B; (B) = cross-section C. confusing! "restoration"

---

## Author Comment (AC1)

*Dear editor.*

Here follow our responses to the review by Thierry Dumont (RC1) of our manuscript submitted to Solid Earth.

Reviews are listed in black - italic while our answers are in blue - plain text.

• *Review of the manuscript* **"Chronology of thrust propagation from an updated tectono-sedimentary framework of the Miocene molasse (Western Alps)",** *Kalifi et al., submitted to Solid Earth, july 2021 Thierry Dumont, CNRS, ISTerre, Université Grenoble Alpes*

**Overview**

*The topics of this manuscript is of major interest, since the Tertiary sedimentation allows to constrain the orogenic propagation in the Alpine foreland. The core of the manuscript consists of integration of new geochronological data (biostratigraphy, chemostratigraphy, magnetostratigraphy) with the existing database in a sequence statigraphic framework, along with a synthesis of available well-log and seismic profiles. Besides this, the authors provide a reappraised structural framework based on existing maps, subsurface information from key seismic profiles, and field overview of some key areas. This reappraisal also benefits from some previously published field sections or even unpublished elements from the geol- alp.com website. These sedimentary and structural synthesis are correlated to propose a dynamic tectono-sedimentary and paleogeographic framework of the forward propagation of Alpine orogeny since Oligocene times.*

*This work is clearly worth of publication, provided it takes into consideration the comments listed below. The most reliable and solid input is the chronostratigraphical synthesis, and the important information brought by field sections and wells/profiles analysis. I have more reservations about the structural synthesis, which lacks kinematic data about brittle deformation (thrusts, faults), folds analysis, ductile deformation (Bornes) and transport directions, which also lacks 3D maps analysis, and which attempts anyway to conclude about stress evolution and chronology of thrusting. Thus, some conclusions appear overinterpreted, such as the complete allochtony of the northern Subalpine massifs, or the attribution of an Oligocene age to the earliest identified thrust (the paper is furthermore devoted to Miocene). From geodynamic point of view, the demonstration and time-space quantification of the forward propagation of the Alpine front since early-middle Miocene is convincing, although the geodynamic and structural inheritance and specificity of the Oligocene phase, along with older inherited structures (Hercynian trends), could have been better considered.*

We are happy to see that reviewer #1 considers that our work clearly worth publication. It is clear that our main contribution are the chronologic constraints that we bring and we understand the concern about some of our structural conclusions, most of them not being the main focus of the paper. We answer in more details below to that concern.

**General comments**

*Organisation of the manuscript is correct but the "Geological setting" and the "overall structure.." § partly overlap concerning the description of main thrusts, their description in §2 could be simplified as both are describing fig. 1B; lines 115 and following partly duplicate § 4.3 (i.e. "the southern prolongation of this fault is contentious" found line 120 and line 348).*

The Geological setting (§2) is a brief summary of what is known on the structure of the area, while the § 4.3 goes deeper into the details. We slightly changed the text to keep the general picture for §2 and the details for § 4.3.

*The structural descriptions of §4.3 are long and tedious, even the 4.3.6 "summary" trying to justify the options choosen in the cross-sections. Could be better organised and shortened.*

§ 4.3 brings a lot of descriptions of local structures and comparisons with previous work in order to discuss the fault zones geometry. This may appear long and tedious. We have significantly rewritten that section, and hope that it is easier to follow.

*Stratigraphy, sedimentology*

*This is of course the main input of the paper, thanks to field sections and synthesis of borehole data and seismic profiles. Synthetic sections provided in appendix are original and essential. The chronostatigraphic integration of different methods seems solid, and provides an essential framework to analyse the tectonosedimentary features.*

*Structures and deformation*

*Despite the structural study refers to "new field data", synthetic presentation of these new data is lacking (i.e. line 389 "210 stations.." without location map nor data synthesis). Kinematics and structural chronology are tentatively deduced from maps analysis (fold and thrust trends), which is not suitable for proper identification of stress directions and discrimination of deformation phases. There is confusion between finite deformation (folds, thrusts) and "shortening phase", suggesting that paleostress can be inferred from the present structural trends, which is abusive. Thrusts and folds are oblique (eastern thrust in Chartreuse is western in Bauges) and sometimes curved. Such variations could be explained by structural inheritance beneath the foreland, with the possible influence of basement inherited structures oblique to Alpine stress, however this would require a specific microtectonic analysis is to determine paleostress, which obviously is not the main aim of this paper. Nevertheless, such analysis will now be facilitated by the improved chronostratigraphic framework provided by it.*

We have indeed conducted new field work and collected new structural data during the course of this study at more than 730 locations (Most of these data are bedding measurements used to constrain the 3D geometry and the cross-sections. This is clearly stated in chapter 3 "Material and methods" and the location of these stations are given in figure S1, and we have now added a table in the supplementary data (table S6). Such measurements have been used to calculate the fold axis of the Proveyzieux and Sassenage anticlines (Fig. 7). We have not conducted precise paleostress analysis that would require another PhD, and are aware that our approach is approximate and thus give only broad estimates of the directions of compression.

*Some more specific comments about FZ1:*
*The connection of thrust 1 both to the western Bauges and to the eastern Bauges implies that the Bauges and Bornes are regarded as an allochtonous nappe. This extreme opinion should be documented by structural observations in the Eastern Bauges massif, which are lacking. Although deformed, the Jurassic cover near Ugine is not detached but only affected by distributed shear. The autors use the data of Gidon's website (i.e. fig. B1C = "unpublished" http://www.geol- alp.com/h_mt_blanc/_schemas/coupe_Aravis_mtBlanc_4.gif) but Gidon himself does not consider that the sedimentary cover of the Bornes massif is detached, although it looks sheared and deformed. The authors should provide additionnal structural data from their own to support the detachment interpretation, or give a better consideration to ductile deformation.*

We indeed present no new observations for the geometry of that fault zone on the NW flank of Belledonne. We rely on previously proposed observations and interpretation (i.e. Deville et al., 1994; Barfety and Gidon, 1996; Barféty and Barbier, 1983; Doudoux et al., 1982, 1999). We use this to suggest that the situation more precisely described in the South (Moucherotte) probably extends further north as already proposed by others (i.e.; Lacassin et al., 1990; Menard and Thouvenot, 1987). Such geodynamic evolution makes sense with respect to our sedimentological data, but we do not pretend that the debate is definitively closed.

*Thrust 1 shows a strong lateral increase in amplitude from S to N. It seems that accomodation is increasing northwards during the deposition of sequences S2 and S3, maybe the correlation between both should be underlined.*

We have not underlined this correlation because thrust 1 is probably not active during deposition of sequence S2 and S3.

*The onset of activity of FZ1 ("phase 1) is proposed as early Oligocene (5.1.1, fig. 16). I do not understand the argument for this? 1) eastward thickening of the Oligocene sequence in profiles 10B may be a flexural response related to more internal structures than FZ1, 2) Oligocene series are transported in the hangingwall of FZ1 (Bauges), 3) in the southern termination of FZ1 around Grenoble, there is absolutely no evidence for Oligocene activity, the thrust is overlying Aquitanian/Burdigalian, 4) early Oligocene corresponds to the activation of the Penninic thrust, further deformed by FZ1. This opinion must be better discussed and justified, and the "P1" bar in dashed line for Oligocene.*

We do agree that it remains some questions about the FZ1 activity and the duration of the first deformation phase (P1), and especially its timing of initiation. On fig.16 we therefore have dashed the FZ1 line for the pre 21 Ma history.

1) Yes, we do agree that it is another possibility, which is clearly mentioned in the paper (end of the paragraph 5.1.1).
2) and 4) In the Bauges area, it depends on which Oligocene we are speaking about. In the Rumilly syncline, the Oligocene is mostly represented by Upper Oligocene and also Lower Aquitanian deposits. To the east, in the Bauges massif, east of the Entreverne thrust (E, Fig. 1), these deposits are absent or very thin, while Lower Oligocene and Eocene deposits are dominant (these deposits occurred during the Penninic thrust activity). Between these two domains, there is the "Les deserts", "Leschaux" and "Les Aillons" syncline showing both Eocene/Lower Oligocene deposits and Upper Oligocene/Aquitanian deposits (phD Kazo, 1975). We think that the FZ1 history was complex and most probably propagating during Upper Oligocene, from the Entreverne thrust to the frontal thrust called FZ1 in this paper but we agree that more detailed work is needed to constrain that history as now clearly expressed at the end of paragraph 5.1.1. Moreover, the first marine transgression never reached the domains east to the FZ1 suggesting that the FZ1 constituted a morpho-structural barrier at least since 21.0 Ma. In this scenario, the presence of Oligocene deposits in the hangingwall of the FZ1 is possible and not contradictory with the scenario proposed in this paper.
3) Around Grenoble, in the footwall of the FZ1, only few meters or tens of meters of Oligocene outcrops (Col de la Charmette). Above, these the Miocene is atributed to the sequence S1 (21-18 Ma) while upper Burdigalian sequences (S2, S3, 18-15 Ma) were not dated but probably present according the sequence stratigraphy interpretations (Fig. A2). The deformation of Lower Miocene units (21-18Ma) in the footwall of the FZ1 is compatible with an activity of FZ1 between 21Ma and 18Ma.

*Other ZF:*
*The connexion from the Subalpine front to the Jura debatable:*
*The S-N continuity of thrust zones 2-3 remain a matter fo debate even with the help of seismic profiles (i.e. FZ2 discussion p. 22)*

Indeed, the continuity of FZ2 is not so clear north of St Laurent Du Pont. We discuss that point in details and then use 2a in the south and 2b in the north.

The continuity of FZ3 was in fact not discussed at all. We have added that sentence: "According to the published geological maps the Chailles anticline appears to connect in the north with the Tournier anticline (TO) (Fig. 1B), suggesting that the FZ3 could connect with the Jura internal thrust (fIJu) (Fig. 1B)." and a question mark on Fig. 1B and Fig. 16.

*The offset of FZ4 is much greater to the S (Royans) and tends to lower in front of N Vercors and Chartreuse (fig.9), this makes its northern cotinuity towards Jura questionnable.*
*The extent of FZ5 to the N is not well constrained (line 522)*

We do agree that the northern continuity of FZ5 is not well constrained and we have added question mark on the maps (Fig. 1 and Fig.16). The FZ4 continuity is better constrained, at least until the 91CHA1-91CHA2 profile to the North. However, we do agree that without outcrops and seismic profile, its continuity with the southern Jura transfer fault is difficult and questionable. We have added a question mark on the maps (Fig. 1 and Fig.16) north to the 91 CHA1-91CHA2 profile

*Moreover, the connexion of FZ2-FZ3 towards Jura + the interpretative roots of FZ2-FZ3 beneath Belledonne would imply that the Jura thrusts are rooted beneath N Belledonne, that is 150km further SE.*

We assume this hypothesis and the S-N continuity of thrust zones 2-3 because North to the Chartreuse massif, the triassic decollement level is much more efficient. Triassic evaporites thickness increase to the North (from north to south; ~500m in the HU2 well log; ~200m in the LTA1 well log; ~50m of sandstones and clays in the PA-1 well log)

*More generally, the boundaries of the structural domains appear to be defined and chosen "à priori" by map synthesis (fig. 1B, §2), so that the final identification of thrust activity is partly a circular argument.*

We are not sure to understand how the fact that the structural domains were defined from map synthesis would render the identification of the thrust activity a circular argument. As a matter of fact the sedimentologic domains have been defined independently from the fault zones which we think is a strong argument in favour of a structural control of the sedimentology.

*Geodynamics*

*Oligocene is a very specific period from geodynamic point of view (west European rift system, Ligurian sea rifting) and the study area is located at the hinge between the Alpine orogen and the rift system. The inheritance of Oligocene structures and paleogeography could be better introduced. More specifically, I think that the "forebulge" interpretation put forward in section 5 and fig. 17 is, to me, somehow model- driven. The study area is large, and it would be worth to distinguish the flexural foreland situation in front of the Swiss molasse basin to the NE, from the Rhone valley rift to the SW. While a forbulge uplift seems reasonable in the former case (section A), the Royans Oligocene paleorelief is more probably related with the large-scale half-graben structure nicely illustrated in fig. 9D, whose Miocene reactivation of the eastern part localized FZ4-FZ5. I would not identify this structure as a forebulge.*

We do agree that Oligocene has been a time of creation of normal faults from Alsace to the Rhône valley, with some of them within the alpine system. These may have created paleo-reliefs latter affected by the alpine compression. We do not think however that these normal faults have been systematically reactivated during the compression. Concerning the Bas-

Dauphiné basin, a Seravalian depocenter (sequence 5 14-12 Ma) exist in the footwall of the FZ5 (Fig. 15C), and we suggest it is related to the activation of that fault zone. It is true however that such depocenter could have been created by another mechanism, for example a normal fault. However, this is unlikely as we have no other evidence of normal faulting at that time. We have changed the text as follow: "In the FZ5 footwall the presence of a sequences S5a-S5b depocenter  (~14.0 to ~12.0 Ma) (Fig. 15C) suggests  a continuous westward migration of the depocenters (Fig. 17), that could be controlled by the FZ5 activity at that time. This hypothesis is consistent with the general trend …", and we have removed the term "forebulge" from Figure 17.

***Detail points***

*49-51 N-NW directed from Eocene to earliest Oligocene, driven by Adria-Europe convergence, then W directed motion driven by extrusion of the internal W Alps, leading to the radially propagating arc.* The sentence has been modified.

*59 "..Vercors and Chartreuse.."* Done

*60 "..NNW-SSE trending Miocene thrusts.."*          Done

*63 "..date from the late Burdigalian.."*        Done

*65 "Further south, .."*           Done

*73 "..well-logs and field sections."* The sentence has been modified.

*77 "as a response to the western propagation of the Alpine orogen during the Miocene."*        Done

*81 also quote De Graciansky, P.C. de, Roberts D.G., Tricart P. (2011), The Western Alps, from rift to passive margin to orogenic belt, an intergated overview. Developments in Earth Surface Processes, 14, Elsevier, 398p., ISBN 9780444537249*        Done

*103 "..the westward progressive migration.."?*        Done

*105 fluvial deposits?*        Done

*108 "..folds and thrusts affecting the sedimentary cover, trending NNW-SSE.." some folds are ~NS? "last WNW-ESE shortening phase" is overinterpreted, paleostress cannot be assumed to be perpendicular to present fold trends, nor can be determined along balanced cross sections whose orientation was choosen à priori.*
Here we refer to published work. Furthermore, if the exact direction of compression cannot be known precisely the apparent direction of shortening is approximately perpendicular to the fold axis.

*112 please locate (from N to S of what?)*    The sentence has been changed to clarify that point.

*120 a more recent ref. about the Moucherotte thrust is still missing (Donzeau, Gamond & Mugnier, CRAS 1993 vol 317 p. 1675)* The reference was added

*146 split the sentence after Gidon, 1964*    Done

*150 "..the Jura fold belt is progressively widening northwards."*        We Changed the sentence to "…Jura synclines widen progressively northwards. »

*151 "..anticlines that develop in the hangingwall of blind thrust.."*    Done

*160 also quote Lickorish et al. (2002) GSA Bulletin 114, 9, p. 1089 168 "..* Done

*168 from 35 well-log and outcrop sections.."* We kept the original formulation.

*171 scale resolution?*        Done

*331 "..in the footwall of the Penninic thrust.."*        Done

*347 "..thrust over.."*        Done

*349 not sure that "a saddle of the folds" is correct..* We now use "fold saddle". We hope this his correct.

*357 "new field observation" should require structural measurements, only landscape views are presented.* If the figure 6 mostly shows landscape views our study is based on observations including measurements at more than 200 stations. Also see related general comment above.

*358 "suggest" incorrect*    We changed the sentence to "We combine new field observations with the published geological maps and other publications to produce a new structural map…"

*361 does it mean that the Sassenage anticline is partly older than Miocene? to be demonstrated, fig. 6B does not show anything*   This is a simple description and it does not imply that the Miocene deposits are unconformable. This is what can be seen on Fig. 6B. We have added "conformably".

*361 "On each side.."*   Done

*362 how can you determine fold trends so precisely without structural data? not visible in fig. 7a.* We indeed have several structural measurements that are now plotted in Fig. 7a, and other ones none plotted. All structural measurements are now given in Table S6 and stereograms used for calculation of fold trends are presented in Fig. 7.

*364 a better argument than the map trace would be a view of 3D geological map.* Yes, we have added Fig. B1 that shows 3D views of the geological map of Fig. 7.

*365 this is already proposed by Gidon 1995, fig. 1, must be quoted*   Done

*367 is this left-lateral displacement necessary considering the dip of the Neron-Moucherotte thrust and the topography? 3D view would be useful*   We now also refer to the new Fig. B1

*389 what means "..at 210 stations"? where are the location/data? should be removed if it is not presented. I do not find this detailed map of the Moucherotte thrust. Would require at least 3D map view for discussion.*

It means that we have performed field work at 210 stations with landscape observations (some of which are shown in Fig. 6), and / or stratification and fault measurements. Only the stratification measurements are plotted on Figure 7. That figure is intended to be the detailed map of the Moucherotte Thrust, and we have added Fig. B1 for 3D views. All structural measurements are now given in Table S6.

*391 "..thrust above the Miocene on top of.."*   Done. This part of the text has been changed.

*395 this thrust at the base of Comboire cliff is not the main Moucherotte thrust because the footwall rocks are Berriasian, vhereas the footwall of the main thrust should be younger (Valanginian, see Gidon, geol-alp website, page Comboire, and quote it). The argument is difficult to understand from your text.* This part of the text has been changed and the cross-section has been modified.

*401 ".. overlain by.."*   Done

*402 fig. 6C? "..the lower Cretaceous strata are steeply dipping eastwards"*   Done

*403 text description very difficult to follow* This part of the text has been changed.

*405 "..in the footwall of the thrust."*   Done

*414 "extend"*   Done

*417 "..but dips locally to the west, strongly suggests that.."*   This part of the text has been changed.

*419 in contradiction with the trace of the "Corenc tectonic window"..* Not sure to understand this comment. There is indeed a large uncertainty of the precise mapping in the Grenoble area. The precise mapping of the thrust(s) and potential window(s) and klippen(s) is hiden by the Quaternary cover. We have added a question mark in that zone to stress out that the mapping is still unclear. Furthermore, there is a large anticline (Ecoutoux Anticline) that appears to be a late fold, but that is distinct from the Conest one. Our text was ambiguous to this respect. We have changed the text and hope it is now clearer.

*420 no, "External Belledonne" is basement, here this is cover* The formulation has been changed.

*424-425 the displacement along the "Corenc-Jalla" thrust seems much lower compared to Neron.* Yes, possibly. We still have to work on that aspect. The corenc Jalla is possibly only one branch of the Neron thrust. But this would not change our general conclusion.

*431 I do not understand this sentence, please clarify.* The sentence has been changed to be clearer.

*435 there seem to be confusion between a "thrust" and a shear zone, what is the thickness of the folded section?* It is clearly indicated that this a shear zone, not a single thrust. We have added an estimate of the thickness of the thrust zone.

*438 Bellahsen et al do not figurate a complete detachment across the Bauges massif.* Well, if it is true that it is not clear on all figures of Bellahsen et al., 2014 but its figure 9 depicts in red "Thrusts of

the "Aravis-Granier" units" and the legend states:" Finally, in red, we have represented the thrusts that correspond to the detached part of the ECM cover; i.e., upper part of the Mont Joly unit for example. In other words, along the Préalpes-Mont Blanc section (NE section), the whole Mesozoic cover of the Mont Blanc is represented by the Morcles nappe, while along the Bornes section (SW Mont Blanc), the Mont Joly (French Morcles) is only made of the lower Jurassic layers. The rest of the cover is displaced and composes the "Aravis-Granier" unit ».

*446 once again structural data should be synthetised in stereograms, not as individual numeric values in the text.* A stereo diagram has been added to Fig. 8B

*450 may be this can be interpreted as southward decrease in thrust displacement as for FZ1?* Here we speak of FZ2. At this stage of the text we do not aim to discuss the amount of motion, only the map trace.

*453 veers? sentence unclear* We now use "turns" that is maybe more clear.

*455 this is the classical view from the Grenoble 1/50000 sheet and Gidon papers, see also Dumont & SPIA, 2020, Geol Soc London Spec Publ 486 doi:10.1144/SP486-2019-92.* The two references have been added.

*470 "Its interpretation suggests that.." Complicated discussion, it appears that the S-N continuity of the individual thrust zones is interpretative and debatable, even with the help of seismic profiles.* We agree and now use "suggests".

*485 not found the OU thrust in fig. 9B, inherited from Deville 2021? check consistency: Outherans/Outeran, SA anticline not on fig.10, FZ1 text is 1 on fig.10.* We have now located The OU thrust on Fig. 9B. The correct spelling is Outherans. No mention of the SA anticline in this sentence. We checked that the CS anticline appears both in Fig. 1 and 10.

*489 why "west-dipping GF"? seems to be E-dipping everywhere* Yes this is "east-dipping"

*496 I do not understand, the potential connection is not visible on the profiles of fig.10.* This is the point so no definite option can be reached.

*504 "..by steep west-dipping.."* Done

*509 "..offset is estimated to.."* Done

*508-511 major thrust .. (FZ3) / the main fault (3) : be consistent in the text description* Done

*516 offset along 4 + 4a on line 82... looks greater than 650m (~1,5km?)* We agree, it was a mistake, and we have calculated 1.24km

*526-530 this presentation is a bit oversimplified, both thrust traces and fold trends vary between NS and NE-SW, and even NW-SE following the interpretations about the FZ continuity in Jura.* We now say "mostly trend".

*534 further north what? BZN1 quite impossible to locate on fig.1, too small. What is the importance of this triple occurrence of Js? ("duplicated three times" means that Js occurs 6 times?).* The sentence has been changed.

*535 "an uplifted compartment" where?* The sentence was modified as follows: "Seismic imaging suggests the presence of a thrust uplifting the Brizon basement high (Le Guellec et al., 1990).". The Brizon basement high was added on Fig.11 A, B, C, D.

*543 quote fig. 18 or remove?* We now quote Fig. 18.

*545 "Klippe" is just a result of erosion, it does not have any implication in terms of displacement or location of "roots"* The sentence has been changed.

*550 .. or simply that Subalpine and Jura thrusts cannot be individually connected?* This is always a possibility but it is not the one envisaged in the literature.

*552 also quote Lickorish et al., 2006; Deville 2021.* Do you mean Lickorish et al., 2002? We now cite Deville 2021.

*555 veer?* We now use "turn"

*566-572 this discussion lacks kinematic data, in the absence of which it is not possible to decipher the relative chronology between thrusts and transfer/strike-slip faults. Similarly, the orientation of compression cannot be constrained from maps only. Do the southern Jura thrusts imply a NE-SW compression?* It is right that the direction of the folds cannot be used simply and systematically do deduce the direction of shortening. However it is also true that these folds and strike-slip faults are all compatible with an ESE-WNW compression.

*580 "In the footwall of FZ1.."* Done

*587 "..in the footwall.."* Done

*588 ".. the deposition of Oligo-Aquitanian sediments"* We keep the original formulation.

*591 "..both Oligocene and early Miocene sediments.."* Done

*683 "are proposed using the.."* Done

*718 ".. while the regressive systems tract.."* Done

*723 (and elsewhere) "Therefore, the sequences S6 and S7..."* Done

*724 "..is either synchronous or sequences .."* Done

*751 and §5.1.1 there is no direct evidence of the activity of FZ1 as soon as early Oligocene, the Oligocene sedimentary sequence is compatible with a flexural foreland but, as said line 763, the eastern boundary of this basin is unknown. Moreover, Oligocene is transported in the Bauges massif, suggesting that the orogenic front was located further east. This option should be presented as an hypothesis and the phase 1 (P1) on fig. 16 put in dashed line.*

As mentioned in one comment above, we modified the text as follows:

"The upper Oligocene-lower Miocene tectonic front is therefore located east of the Rumilly syncline. Since marine Miocene deposits have never been described east of FZ1, we believe that it already formed a morphostructural barrier during the first marine sequences (S1a-S1b; 21.0 to ~18 Ma, Fig. 4B, C) and was thus active at least between ~21.0 and 18 Ma, and possibly as early as 29±2 Ma. However, in the early stage, the active front could as well have been located east of FZ1, for example along the Entrevernes thrust [E] (Fig. 1B) or even further east, and other arguments are still needed to decipher the ante-21.0 Ma structural history.". The possible pre 21 Ma activity of FZ1 now appears as a dashed line with question marks on fig.16.

*792 what means « concordant on the folded Urgonian », is Miocene folded with Urgonian ?* Yes. We have changed the sentence to make that point clearer.

*797 the wavelength of flexural subsidence is several hundred km , consistent with the lithospheric thickness, this process can hardly be involved for a depocenter in the footwall of an individual thrust.*

We think that this process can be involved because in the footwall of the orogenic front, two process are adding up: (i) long-wavelength lithospheric deflection in response to subduction process, and (ii) the topographic loads in response to the growing orogenic front. Adding these two processes, the long-wavelength lithospheric deflection curve is accentuated near the footwall of the orogenic front, and the maximum of subsidence is recorded in the footwall of the orogenic front (DeCelles and Giles, 1996; Schlunegger and Kissling, 2015).

*803 hypothesis* Done

*805 "..the thrusts that are interpreted to root beneath the Belledonne massif.."* Done

*806 uncorrect sentence (and the S6 sealing...)* The sentence has been changed.

*807 "end of phase 2" is time, "in the subalpine massifs" is location, unclear.* The sentence has been changed.

*809 "..are overthrust by the Jura units.."* Done

*810 posterior younger than..* Done

*812 tectonically-controlled..* Done

*813 "..even perhaps coeval with the deposition of S7 continental sediments"* Done

*829 "..suggets that thrust activity stopped after the deposition of S5. There, the uppermost marine.."* Done

*834 the link between the arguments presented since line 828 and "activation of out-of-sequence thrusts in an internal position" is unclear. Moreover, such structures should produce subsidence in their footwall, not uplift; and isostatic rebound is mentioned line 833. So please clarify your interpretation of a "tectonic event" involving "lithospheric processes", it is now clear that the recent uplift of the W Alps is not directly linked with plates convergence, and is probably not "tectonic".*

This part of the discussion has been changed (new paragraph 5.3)

This part of the text now reads: "The following structural history however strongly differ between the north and the south: at ~13 Ma in the Aar the deformation front appears to have quicly migrated more than 50 km to the NW to form the Jura, while it ended at ~12 Ma in front of central Belledonne. However, some observations suggest that the Bas Dauphiné experienced uplift after 12Ma. In the western and southern parts of the Bas-Dauphiné basin and the Crest basin (respectively, J, K, L zones, Fig. 3) the Miocene final sea retreat is recorded during deposition of sequence S7 (~10 Ma, Fig. S13, 14, 15, 16). The absence of marine deposits during S8 (~9,5 to ~8 Ma) is unexpected, as it corresponds to an eustatic transgression corresponding to a higher sea level (+40 m) than that of sequence S7 (+5-10 m, Miller et al., 2005, Fig. 4B). This implies that the ~10 Ma Miocene sea-retreat was induced by a basin-scale event. In the north-eastern part of the Bas-Dauphiné basin (F, H zones, Fig. 5), the uppermost marine deposits (sequence S6 regressive deposits, ~11 Ma, Fig. S10, 12) outcrop today at elevation of ~350 m.a.s.l. (Fig. 14B; Fig. 15A). The sequence S6 transgression involved a +25 m sea-level rise which suggests a post-sequence S6 (12 Ma) minimum uplift of ca. 325 m. This is in agreement with Deville et al. (1994)'s observations implyinged a post-Langhian uplift. These authors interpreted this uplift as the result of a crustal thickening due to a crustal imbrication under the molasse basins, implying the activation of out-of-sequence thrusts in an internal position of the subalpine massifs. However, these thrusts have not been clearly indentified. Another possible interpretations are that this uplift would be link to the activation of late backthrusts such as those described along section C (bkt1 & bkt2, Fig. 11), or to a deeper process."

*855 backthrusts are compatible with thrusting, they do not need another tectonic phase?* They do not need to, but it is our interpretation. The sentence has been changed to clarify that point.

*875 the onset of phase 1 as soon as early Oligocene is not well constrained, must be given as hypothetical.* Done

*875-880 there is an overlap between phase 1 and 2? (17,8Ma+/-?? - 18,05+/-0,25Ma), better use Early Burdigalian line 875?* From our data there is possibly a slight overlap, but it is beyond our temporal resolution. So, we will not discuss this in details but keep the numeric ages for comparison.

*881 the "Belledonne basal thrust fault" is actually unknown, no outcrop, no seismic, no borehole. Must be presented as an interpretation.*

We now present this as an interpretation.

*899 "..recorded after the deposition of S5"* Done

*901-903 I do not understand which tectonic structure could be involved to explain this uplift by tectonic processes in Bas Dauphiné? there is no evidence of shortening, and the situation is different from the N Alpine foreland.*

See answer to comment on line 834 above

**Figures**

*Figures are generally of good quality, and contain a lot of information, but will be hard to read on a paper copy.*

*Figs 4 & 5 dense but clear, this is the core of the data*

*Figure 6: A and B are not very clear because of backlight* Modified
*a 3d view of the geological map draped over DEM would perhaps give a better overview.*
*We now Provide Fig. B1*
*Picture H can be processed to get better colours* Done

*Fig. 7: Why are the cross-sections named both with numbers (1-6) and letters (B-G)? confusing. I do not understand the trace of the east part of FZ1 outcropping both W of Corenc and E of Grenoble, not compatible with cross sections 1 & 2.* Only letters of sections are now used. The FZ1 trace east of Grenoble was removed with a (?).

*Fig. 9: are the interpretation based on Deville 2021 or were the lines completely reinterpreted? must be clarified.* The lines were completely reinterpreted and it is clarified in the Methods. It will be also clarified in the Fig.9 caption

*Fig. 11: I remain reluctant considering the FZ1 detachement over Belledonne in section A, and I totally disagree with its trace above Belledonne in section D: the normal fault offset is Jurassic as demonstrated in the 80s (La Mure fault) as the Liassic series have very different thickness on both sides (Lemoine et al., 1986). Must be changed.*

We do agree that the eastern part of our cross-sections B and C are questionable and that of section D was not accurate, as it is clearly not the focus of the manuscript. Question marks have been added to the "Accident median" (AM) and the La Mure fault now clearly appears as a Normal fault (pink) in section D.

*Fig. 13: sections identified using both n° and letters (3 = A, 4 = B etc.), keep only n°?* Only n° of sections are now used

*Fig. 15 please refer to the location of transects on a map* It is refered in Fig.3. The reference was added in the caption of Fig. 15

*Fig. 17: the activation of FZ1 to the east is not consistent with the occurrence of Oligocene deposition in the futuer Bauges massif, that is in the hangingwall of the thrust. The activation of the Belledonne thrust as soon as ~20Ma is in contradiction with the exhumation ages which are younger.*

See answer to point 2) and 4) above. FZ1 fault history was possibly complex and most probably propagationg during Upper Oligocene from the Entreverne thrust to the frontal thrust called FZ1 in this paper. In this scenario, the presence of Oligocene deposits in the hangingwall of the FZ1 is possible and not contradictory with the activation of the Belledonne basal thrust. The exhumation age of Belledone is not yet precisely constrained and some new data suggest that it could be older than previously thougth, but this point is beyond the scope of this paper.

*Fig. 18: (A) = cross-section B; (B) = cross-section C. confusing! "restoration"* We think that there is no solution to this issue

---

## Author Comment (AC2)

Solid Earth Discuss., referee comment RC2
https://doi.org/10.5194/se-2021-46-RC2, 2021
**Comment on se-2021-46**

Here follow our responses to the review by Fritz Schlunegger (RC2) of our manuscript submitted to Solid Earth.

Reviews are listed in black - italic while our answers are in blue - plain text.

*Fritz Schlunegger (Referee)*

*Referee comment on "Chronology of thrust propagation from an updated tectonosedimentary framework of the Miocene molasse (western Alps)" by Amir Kalifi et al., Solid*
*Earth Discuss., https://doi.org/10.5194/se-2021-46-RC2, 2021*

*Dear Authors, dear Editor*

*It is usually a significant challenge to intergrate various sources of data into a comprehensive and conclusive framework. This paper is an excellent example of how this can be achieved. I congratulate the authors for their work, which I enjoyed reading. This has been very well done!*

Thank you

*The material is presented in such a way that the reader can follow the way of how the authors reach their conclusions, and why. The readers are given access to a wealth of material that has been integrated in this manuscript. Therefore, from a scientific point of view, this work is very solid, reproducible and should be published.*
*What has not been fully clear to me is the separation of previously published data (in the authors' Sedimentology paper) and their original contribution presented in this work. In this regard, chapter 3 (Materials and Methods) should be more specific and clear.*

The sedimentology paper (Kalifi et al., 2020) aim was the description and interpretation of the facies and how they evolve in space and time. For this, sedimentological sections 4, 5, 13, 16 and 22 were presented in details, together with 57 Strontium ages which were published without stable isotopes results. In this paper we present 30 more sedimentological sections and ad 72 new Strontium ages to calibrate the log sections, together with biostratigraphy and magnetostratigraphy age constraints. Some sequence stratigraphy interpretations thus slightly evolved.

We modified the chapter 3 as follow:

Lines 208-217: "Sedimentological and stratigraphical analyses were conducted from 35 well-outcropping sections of the Miocene Molasse deposits (sections 4, 5, 13, 16, 22 are detailed in Kalifi et

al., 2020), and from partially preserved sections (<40m) outcropping in adjacent localities. Sedimentary successions, up to 1050 m-thick, were logged at the decimeter (dm) to meter (m) scale resolution in the field. Using the combined analyses of textural characteristics, clastic and biogenic components, bed thickness, bed organization and geometry, sedimentary structures and paleocurrent measurements, 25 facies grouped into 11 facies associations (FA) were previously defined by Kalifi et al. (2020). Building on these results and using the same methodology, depositional sequences were identified based on facies associations evolution and the main stratigraphical surfaces (Embry, 1993, 1995). Depositional sequences were identified, using Posamentier and Allen (1999) methodology on spontaneous potential (SP) and gamma-ray logs (GR) data from 28 well-logs located in the Bas-Dauphiné basin."

Line 226: "To the 57 samples published by Kalifi et al. (2020) we ad 72 new samples (Table S1)."

In table S1 samples published in Kalifi et al. (2020) are now denoted by asterisks (*).

It is true that chronological data for the Burdigalian is sparse for the Molasse deposits. However, the authors might have a look at the magnetostratigraphic work done at the Univ. Bern c. 25 years ago where some terrestrial sections of Burdigalian age have been calibrated through magnetostratigraphy and mammal biostratigraphy (Schlunegger et al., 1996, Eclogae Geol. Helv., Kempf et al., 1997, Int. J. Earth Sci.; Strunck and Matter, 2002, Eclogae Geol. Helv.). Therefore, the first sentences of the Abstract need to be tuned town. I understand that the authors refer to the Western part of the NAFB and the Alpine orogen, because it is not true that the chronological records are poor for the Swiss, German and Austrian segments of the NAFB (perhaps see also Hülscher et al., 2019, Front. Earth Sci.)

*The first sentence of the abstract has been modified as follows:*

*"After more than a century of research, the chronology of the deformation of the external part of the __western__ Alpine belt __(France)__ is still controversial for the Miocene epoch"*

As a final, but not mandatory aspect, I think it would be worth while placing the sedimentary history of the western part of the NAFB into a broader context, if possible. In particular, following Berger (2005; Int. J. Earth Sci.) and Ford and Lickorish (2004; Geol. Soc. London Spec. Publ.), the pre 20 Ma sediments in the western part of the NAFB are characterized by gypsiferous marls, freshwater carbonates and paleosoils, suggesting a sedimentary environment that is indicative for a basin margin which opened towards the Swiss, German and Austrian Molasse basin that was the depositional sink at that time. After 20 Ma and particularly after 18 Ma, the situation changed as the dispersal direction became reversed and as sediment was routed from the Eastern Alps and the Bohemian massif through the German and Swiss Molasse basins and finally to the French part of the NAFB, which started to take the role as a depositional sink. Interestingly, this is the time when active deformation at the orogen front started, as documented in this work, while thrust front activity came to a halt in the Austrian basin.

Another paper, which is in review in Geol Soc Sp. Pub., is dedicated to the paleogeographical evolution of the western alpine foreland basin. However, we do agree that we can briefly compare the timing of the tectonic phases from our results with those of the neighbouring molassic basins (Swiss and the Rhodanian). A new paragraph (5.3) was integrated and is

called "5.3: Comparison of deformation phases affecting the Miocene molasses in western Alps".

Finally, there are a couple of typos to be revised (my review might also contain typos, for which I apologize):
Line 184: samples collected in the field (not on the field) Done
Lines 189 and 190: The terms 'comprised' sounds odd to my in this context. Done ("Comprised" not essential, has been deleted)
Line 232: outlier samples and not outliers samples Done
Line 267: The term 'allocated' sounds odd to me in this context. Done, replaced by "provided"
Line 394: The marls did not deposited.... -> The marls were not deposited. Done. The sentence has been changed (line 505).
Line 396: They rather deposited....(they deposited what?) -> They rather accumulated Done, the sentence has been changed to:

"These marls, previously mapped as Jurassic (Vif geological map; Barféty et al., 1967), are rather Early Cretaceous in age based on the occurrence of Berriasella (Gidon, 2020a)."
Line 401: a boxed anticline overlyied by -> overlain by. Done
Line 446: the faults strike N3, 40°E -> something is missing @N3 Done: N3°, 40°E
Line 566: I could not find the South Jura transfer zone on a map. The name was added in Fig. 1
Line 570: I could not find the left-lateral and right-lateral faults The left-lateral faults are in violet and the right-lateral faults are in blue, as it is mentioned in the legend of Fig.1
Lines 583 and 584: I guess that the thicknesses of 1838 m and 1716 m are taken from a seismic line, which will have their uncertainties. If correct, the precisions given here (to the meters) need to be tuned down.

These datas are from well log datas:

"Indeed, these deposits are ~200 m thick west of the Rumilly syncline (Fig. 9C, D) (Enay et al., 1970; Gidon, 1970b), while to the east, they reach 1838 m between the footwall of the FZ1 and the hangingwall of the SAL fault (**SLV2 well data**, Fig. 10A, B), and 1716 m at the footwall of the SAL fault (**SV-101 well data**, Fig. 10A, D)."
Line 600: Firstly -> First, then second (not secondly), and then third (not thirdly) Done
Line 629: use a different term than 'brutally' (perhaps appropriate for a movie, but not really in a scientific article) Done, replaced by "sharply"
Line 632: A thickening can also be associated with a backstepping of depocenters (in case where sediment supply is lower than formation of accommodation space). Therefore, the inference that a rapid accumulation of sediment implies a depocenter migration is only correct if the sedimentary facies is considered as well. Please adjust accordingly.

Done, modified as follows:

"This firmly demonstrates that a depocenter localized close to section 4 (Fig. 13) appeared during S2a. Subsequently, the thickest accumulation of the following sequence (S2b) lies further west, at the Forezan locality (275 m, section 5, Fig. 13). This lateral variation of the thickness is associated with significant lateral facies variation characterized by a dominance of proximal marine deposits to the east (950-1015 m, section 4, Fig. 13), while to the west, S2b is mainly represented by distal marine deposits (700-920 m,

section 5, Fig. 13), thereby suggesting a westward migration of the depocenter between sequences S2a and S2b."

Line 636: 'It was never recorded thicker' sounds a bit odd to me. Please rephrase Done, replaced by "while in the Rumilly-Chambéry synclines area (Fig. 13), the sequence S3 was probably much thinner."

Line 691: This interpretation of a complex inherited topography warrants further specifications. Done: "On the western edge of the Bas-Dauphiné basin, the absence of S2a-S2b deposits (to the northwest of PA-1, VAF-2 and MO-3 wells, Fig. 15A, B, C) and the thickness variations of the S3 sequence to the west of the Montmiral high (Fig. 15B, C) are attributed to a complex inherited paleo-topography (Kalifi, 2020) along the Oligocene West European Rift  (Debelmas 1974; Curial 1986; Bergerat 1987; Ziegler 1988, 1990, 1994; Bergerat et al. 1990; Sissingh 2003)."

Line 759: According to DeCelles and Gilles (1996; Basin Research), Schlunegger and Kissling (2015; Nat. Comm.; my apologizes for this self citation), orogenic loads can have different components such as slab loads, topographic loads (both downward directed) and buoyancy forces exerted by a crustal root. Is it possible to be more specific when you talk about 'in response to orogenic load'?

Yes, thank you for the suggestion. It was detailed as follows:

"In a foreland basin, this geometry is consistent with a foredeep depozone located between the poorly subsiding proximal flank of the forebulge and the footwall of the active (tectonic) orogenic front, where the maximum of subsidence is recorded in response to the interplay between topographic loads and long-wavelength lithospheric deflection in response to subduction process (DeCelles and Giles, 1996; Schlunegger and Kissling, 2015)."

Line 773: Why is the deposit illustrated on the photo (the details are hard to see) a seismite? This interpretation is hard to appreciate without further information.

This interval was interpreted as the Facies F25 presented in Kalifi et al. (2020). It consists of a 15-meter-thick interval of tilted/disturbed autochthonous sedimentary layers containing 'Ball and pillow' structures. This unit is laterally continuous at tens of kilometerscale (Same unit was found in the same stratigraphic level, at the Forezan section n°5, 18km to the south). The organization of the autochtonous clasts of various size is chaotic suggesting earthquake-disturbed layers (i.e. seismites) and the lateral continuity at the basin scale indicate strong disturbance events.

Details are now given in lines 605-610 (modified in order to be more accurate): "Second, a 15m-thick interval (390-405m, Fig. 12A), containing disorganized monogenic clasts of various sizes (cm to pluri-m, Fig. 12C) and 'ball and pillow' structures (Fig. 12D), with a pluri-km lateral continuity (also described 18 km to the south, in section 5, at ~380 m, Fig. 2) suggesting an earthquake-disturbed layer (i.e. seismites, F25 of Kalifi et al., 2020)."

Line 833: What is the evidence for a rebound, and a rebound related to which process?

According to Deville et al. 1994, the isostatic uplift is linked with crustal thickening in response to new crustal imbrication under the molasses basin involved by a late active tectonic deformation (maybe the term "rebound" we used wasn't adapted). From Deville et al. 1994 :" (uplift from the Langhian below sea-level to the present mean altitudes ranging between 500-800m, e.g., much higher than the mean altitude of coeval sediments in the Bresse basin; Bergerta et al, 1990). This could be related to the late active tectonic deformation of the foreland. Indeed, a possible interpretation could be the present development, at depth, of a new crustal imbrication that is suggested by the ECORS deep seismic results where a thickening of the lower layered crust appears under the molasses basin (Guellec et al., 1990a). The crustal thickening could be responsible for an isostatic uplift of the foreland. Note also that an active strong uplift is currently taking place in the SE parts of the Jura (Fourniguet, 1977)."

The sentence was modified as follows: "The following structural history however strongly differ between the north and the south: at ~13 Ma in the Aar the deformation front appears to have quicly migrated more than 50 km to the NW to form the Jura, while it ended at ~12 Ma in front of central Belledonne. However, some observations suggest that the Bas Dauphiné experienced uplift after 12Ma. In the western and southern parts of the Bas-Dauphiné basin and the Crest basin (respectively, J, K, L zones, Fig. 3) the Miocene final sea retreat is recorded during deposition of sequence S7 (~10 Ma, Fig. S13, 14, 15, 16). The absence of marine deposits during S8 (~9,5 to ~8 Ma) is unexpected, as it corresponds to an eustatic transgression corresponding to a higher sea level (+40 m) than that of sequence S7 (+5-10 m, Miller et al., 2005, Fig. 4B). This implies that the ~10 Ma Miocene sea-retreat was induced by a basin-scale event. In the north-eastern part of the Bas-Dauphiné basin (F, H zones, Fig. 5), the uppermost marine deposits (sequence S6 regressive deposits, ~11 Ma, Fig. S10, 12) outcrop today at elevation of ~350 m.a.s.l. (Fig. 14B; Fig. 15A). The sequence S6 transgression involved a +25 m sea-level rise which suggests a post-sequence S6 (12 Ma) minimum uplift of ca. 325 m. This is in agreement with Deville et al. (1994)'s observations implyinged a post-Langhian uplift. These authors interpreted this uplift as the result of a crustal thickening due to a crustal imbrication under the molasse basins, implying the activation of out-of-sequence thrusts in an internal position of the subalpine massifs. However, these thrusts have not been clearly identified and it is unclear how such thrusts could induce uplift of the Bas Dauphiné. Other possible interpretations are that this uplift would be linked to the activation of late backthrusts such as those described along section C (bkt1 & bkt2, Fig. 11), or to a yet unclear deeper process."

Line 836: ECMs = external crystalline massifs (please in full) Done

Line 885: Seismite, same as above. See answer above (line 773).

Line 887: 'brutal', same as above Done, replaced by "rapid"

Please do not hesitate to contact me if you have questions on my review.
Sincerely
Fritz Schlunegger, Bern, August 12th

---

## Author Response (AR2)

**Comments to the author**:

**Dear Dr. Kalifi,**

**I concur with Dr. Sue that your manuscript is fit for publication, albeit pending a final revision. I urge you to make sure the syntax of units and references in your figures (including those in the supplementary material) follow those of the article/journal. You will need to double-check your references. I found a few mistakes, and I suspect there are more to be found. Finally, my comments are mostly regarding your illustrations.**

**Best wishes,**

**Arjen Stroeven**

Dear editors,

Thanks for the minor revision. We are pleased to propose you a new version of our manuscript following the review phase. We have responded point by point to the remarks/suggestions/corrections (see below).

We have made the necessary changes and corrections in the text ("Kalifi_et_al_manuscript_se-2021-46-review2-correction-track" file). The references were checked and the figures were updated.

Thank you in advance for continuing the submission process,

Yours sincerely

Amir KALIFI

In behalf of all co-authors

**L.93 : « 1943 »** *Corrected*

**Fig. 1 (pg.5) « 45.000 E, 5.000 N, Genève, St-L.a » :** *Corrected* **/ « Transfert fault »** *: Replaced by transfer fault zone*

**L.186 : « 1988 »** *Corrected*

**L.222: « Deville, 2021 »** *Corrected*

**Fig. 2 (pg.10) « Dating techniques; millions of years » :** *Corrected*

**Fig. 3 (pg.12) « 45.000 E, 5.000 N, Genève »** : *Corrected*

**L.301. " Bas-Dauphiné ":** *The consistency within the manuscript and the figures and the Supplement materials were checked*

**L.301."Refer to figure 1 for additional information not mentioned here in the legend or the caption.":** *Added*

**Fig. 4 (pg.13) « Uppermost »:** *Corrected*

**L.306: "Paleogeographical":** *The consistency within the manuscript and the Supplement materials were checked*

**L.306: "explain the red arrow":** Red arrow corresponds to the westward-directed

**Fig. 5 (pg.13): "cysts" :** *Corrected*

**Fig. 7 (pg.19): "geological; PALEO-ZOIC" :** *Corrected*

**L.444: "Barféty":** *Corrected*

**Fig.8 (pg.22): "the cross-section lacks a length axis scale.":** *The length scale is in the lower left corner, above the legend.*

**Fig.9 (pg.23): "Quaternary":** *Corrected*

**L.490: "Deville, 2021":** *Corrected*

**L.492: "Outherans (Ou)":** *Corrected*

**Fig.10 (pg.24): "Quaternary, Tertiary, et al., x km":** *Corrected*

**L.497: "Figure":** *Corrected*

**Fig.11 (pg.28): "make depth values negative":** *Corrected*

**L.588: "figure panel descriptions A-E do not match figure panels.":** *Corrected*

**L.588: "give fig#":** *Corrected*

**Fig.12 (pg.30): "Dating/Dating techniques":** *Corrected*

**Fig.13 (pg.32): "paleogeographical, Seismite ":** *Corrected*

**Fig.14 (pg.34): "Add N and E":** *Corrected : "list depth m? Altitude m asl? Explain PS-reverse and "mv":* *Corrected*

**Fig.15 (pg.36): "probably correct but hard to use reference":** *We added "Montmiral high" in Fig. 9C /* **"m a.s.l.; Dating/Dating techniques/ remove "+" in A-C"/ Posamentier and Allen, 1999/ 200 m 2 km"** *: Corrected*

**L.731: "Figure 15":** *Corrected*

**Fig.16 (pg.38): "put white box behind?":** *A box was added*

**Fig.17 (pg.40): "Inherited":** *Corrected /* **"this refers to information in figure 18? If not, use figure caption to explain.":** *Yes*

**L.811: "Figure 17":** *Corrected*

**Fig.18 (pg.43): "separate km and m from values":** *Corrected*

**L.880: "amounts":** *Corrected*

**L1027: "Barféty, all 1996 references are wrong, except in Figure 7":** *Corrected*

**L1029: "arrange before B&G 1996":** *Corrected*

**L1080 to 1089: "not in the text, move to SM / references that are not used in the text (such as this one, but there may be several others) should not appear in this list of references. Where used in the supplementary material, please compile a list of references there.":** *Moved to Supplement materials*

**L1085: "Chevenoy, see Clauzon":** *All references which appears in the Supplement materials are now included within the SM and excluded from the main manuscript if it doesn't appear in it.*

**L1110: "to SM":** *Moved to Supplement materials*

**L1193: "to supplementary material":** *Moved to Supplement materials*

**L1205: "to SM":** *Moved to Supplement materials*

**L1221: "Jeannolin (1985)":** *Moved to Supplement materials*

**Fig. S1 (SM pg2): "wrong reference":** *Corrected*

**L36 (SM): "not "et al." in your list of references. There ARE mistrakes, so please double-check all of your references.":** *Checked*

**L41-42 (SM): "are all the vertical sections in "m"? If so, add to each profile or state in the figure caption.":** *All sections are in m and it was added in figures. (some values are given in m a.s.l. for the equivalence but it is now better differentiated)*

**Fig. S5 (SM pg7): "Dating, Seismite":** *Corrected*

**Fig. S6 (SM pg8): "Dating, Bottomsets":** *Corrected*

**Fig. S7 (SM pg9): "Dating":** *Corrected*

**Fig. S8 & S9 (SM pg10): "Dating":** *Corrected*

**Fig. S10 (SM pg11): "Dating; m a.s.l./ ? correct?":** *Corrected*

**Fig. S11 (SM pg12): "Dating":** *Corrected*

**Fig. S12 (SM pg13): "Dating; m a.s.l.; keep the same style throughout; not italics; (Gigout et al., 1976)":** *Corrected*

**Fig. S13 (SM pg14): "Dating; m a.s.l.; 300 m; 1-5 m; 1000 m; same style throughout":** *Corrected*

**Fig. S14 (SM pg15): "Dating; m a.s.l.; (Gigout et al., 1976)":** *Corrected*

**Fig. S15 (SM pg16): "Dating; m a.s.l.; marine?; (Gigout et al., 1976)":** *Corrected*

**Fig. S16 (SM pg17): "Dating; m a.s.l.; Sables à Unios de M.; references":** *Corrected*

**Fig. S17 (SM pg18): "Cysts, HCS needs explanation, Kalifi et al., 2020":** *Corrected*

**Table S2 (SM pg24): "m a.s.l.; subscript":** *Corrected*

**Table S3 (SM pg30): "m a.s.l.":** *Corrected*

**Table S4 caption (SM pg39): "1998; SEPM chart":** *Corrected*

**Table S5 (SM pg40): "m a.s.l.; rare":** *Corrected*

**Table S6 (SM pg44): "N and E; ":** *Corrected*

**Table S7 (SM pg69): "Uncertainty; use same style":** *Corrected*